# CaLFADS: latent factor analysis of dynamical systems in calcium imaging data

## Abstract

Dynamic latent variable modelling has provided a powerful tool for understanding how populations of neurons compute. For spiking data, such latent variable modelling can treat the data as set of point-processes, due to the fact that spiking dynamics occur on a much faster timescale than the computational dynamics being inferred. In contrast, for other experimental techniques the slow dynamics governing the observed data are similar in timescale to the computational dynamics that researchers want to infer. An example of this is in calcium imaging data, where calcium dynamics can have time timescales on the order of hundreds of milliseconds. As such, the successful application of dynamic latent variable modelling to modalities like calcium imaging data will rest on the ability to disentangle the higher- and shallower-level dynamical systems' contributions to the data. To-date, no techniques have been developed to directly achieve this. Here we solve this problem by extending recent advances using sequential variational autoencoders for dynamic latent variable modelling of neural data. We solve the problem of disentangling higher- and shallower-level dynamics by incorporating a ladder architecture that can infer a hierarchy of dynamical systems. Using some built-in inductive biases for calcium dynamics, we show that we can disentangle calcium flux from the underlying dynamics of neural computation. First, we demonstrate with synthetic calcium data that we can correctly disentangle an underlying Lorenz attractor from calcium dynamics. Next, we show that we can infer appropriate rotational dynamics in spiking data from macaque motor cortex after it has been converted into calcium fluorescence data via a calcium dynamics model. Finally, we show that our method applied to real calcium imaging data from primary visual cortex in mice allows us to infer latent factors that carry salient sensory information about unexpected stimuli. These results demonstrate that variational ladder autoencoders are a promising approach for inferring hierarchical dynamics in experimental settings where the measured variable has its own slow dynamics, such as calcium imaging data. Our new, open source tool, thereby provides the neuroscience community with the ability to apply dynamic latent variable modelling to a wider array of data modalities.

## 1 Introduction

Dynamic latent variable modelling has been a hugely successful approach to understanding the function of neural circuits. For example, it has been used to uncover previously unknown mechanisms for computation in the motor cortex (Churchland et al., 2012; Pandarinath et al., 2018), somatosensory cortex (Wei et al., 2019b), and hippocampus (Chaudhuri et al., 2019). However, the success of this approach is largely limited to datasets where the observed variables have dynamics whose timescales are much faster than the dynamics of the underlying computations. This is the case, for example, with spiking data, where the dynamics governing the generation of individual spikes are much faster than the dynamics of computation across the circuit. This makes it possible to characterise the observed data, e.g. the spiking data, as a set of point-processes that can be used directly for inferring latent variables.

However, many datasets in the life sciences are generated by a hierarchy of dynamical systems, wherein the shallower-level dynamical systems that directly generate the observed data have temporal dynamics whose timescales overlap with that of the deeper-level dynamical system to be inferred.

A clear example of this is *in-vivo* calcium imaging data, which is widely used in neuroscience. Thanks to advances in imaging technology and genetically encoded calcium indicators, calcium imaging enables monitoring of the activity of large populations of genetically targeted neurons in awake behaving animals (Yang & Yuste, 2017; Lin & Schnitzer, 2016). However, calcium imaging introduces an additional layer of a relatively slow dynamical system between the computations occurring in the brain and the measurements that neuroscientists make. This problem is outlined in Figure 1A, in which calcium fluorescence observations, $x$, depend on the state of calcium flux, $z_1$, which is governed by a shallower-level dynamical system with temporal dynamics on the order of hundreds of milliseconds. These dynamics are driven, in part, by perturbations due to spikes, $u_1$, which are themselves governed by a computational state, $z_2$, with a similar timescale in its dynamics to the calcium flux (and which itself can be perturbed independently by unknown inputs, $u_2$, that may also have slow dynamics). Due to the overlap in timescales, it is impossible to identify immediately which components of the calcium fluorescence data are driven by the dynamics of calcium flux, and which are driven by the deeper-level latent dynamics of neural computation. Ideally, neuroscientists would have a method for inferring both the shallower-level calcium dynamics and the deeper-level computational dynamics in order to uncover the hierarchical dynamical systems that generated their data. Such a tool would be significantly benefit the systems neuroscience community.

Currently, these problems are treated separately. For situations where the observed data can be treated as a point process, we have good techniques for inferring the deeper-level dynamics. For example, recent applications of sequential variational autoencoders have seen great success in inferring underlying computations from extracellular spiking data (Pandarinath et al., 2018). This technique, named Latent Factor Analysis of Dynamical Systems (LFADS), has improved neuroscientists' ability to infer underlying neural computations from spiking data, e.g. it has been used to identify latent rotational reaching dynamics and to decode reaching behaviour of macaques and humans with higher fidelity than other techniques.

Although LFADS has significantly advanced our ability to analyze neural data in the form of spike trains, it does not address the problem highlighted above for calcium imaging, wherein the calcium dynamics introduce an additional shallower-level dynamical system whose timescale overlaps with the timescale of neural computation. Theoretically, this problem could be solved independently by first inferring spikes from calcium data, whether by deconvolution (e.g., OASIS) (Friedrich et al., 2017), variational inference (e.g., DeepSpike) (Speiser et al., 2017), dynamic programming (Deneux et al., 2016), or any other method (Berens et al., 2018; Pachitariu et al., 2018; Evans et al., 2019; Wei et al., 2020), and then applying LFADS. However, this approach treats each calcium trace as a completely independent variable when inferring calcium dynamics. This ignores correlations in population activity that inform the separation of calcium dynamics (which are independent of population activity) from computational dynamics (which are not independent of population activity). If this separation is sub-optimal, then inference of the deeper-level system will be impaired.

Here, we address this problem by extending LFADS with a variational ladder autoencoder architecture (Zhao et al., 2017) that folds the calcium dynamics inference into the larger inference problem (Fig. 1B). Our system, CaLFADS, incorporates inductive biases for calcium dynamics and, thanks to the ladder architecture, is able to infer the deeper-level dynamical system better than an approach that treats inference of calcium dynamics and deeper-level dynamics as separate problems. For further discussion on how CaLFADS relates to previous work, see Appendix B.

First, we show using synthetic data that we are able to reconstruct ground-truth latent dynamics from synthetic calcium traces. Next, we apply CaLFADS to spiking data from macaque motor cortex that has been converted into calcium fluorescence traces using a calcium dynamics model. We show that we are able to recover rotational dynamics from this "calcified" data just as LFADS identifies rotational dynamics from spiking data. Finally, we show using real 2-photon calcium imaging data from mouse primary visual cortex that CaLFADS can identify deeper-level latent factors that carry information about unexpected visual stimuli. Altogether, our work shows the benefits of incorporating the calcium dynamics inference procedure into the larger inference problem. It also provides neuroscientists with a new, open-source tool for analyzing calcium imaging data in order to identify deeper-level dynamics. Given the importance of calcium imaging to modern systems neuroscience, we believe that CaLFADS will be a very useful analysis tool for the community. Furthermore, our CaLFADS could be adapted to other neuroscience data modalities, such as fMRI data which, like calcium imaging, comprises both fast deeper-level latent brain dynamics and slower shallower-level blood oxygen measurement dynamics (Ollinger et al., 2001). Beyond this, we believe that our ap-

proach may be applicable to other life sciences domains, for example insurance claim modelling, where better identifying the relatively slow dynamics underlying the observed variables, such as claim submission distributions over time, could improve forecasting accuracy (Schmidt, 2014).

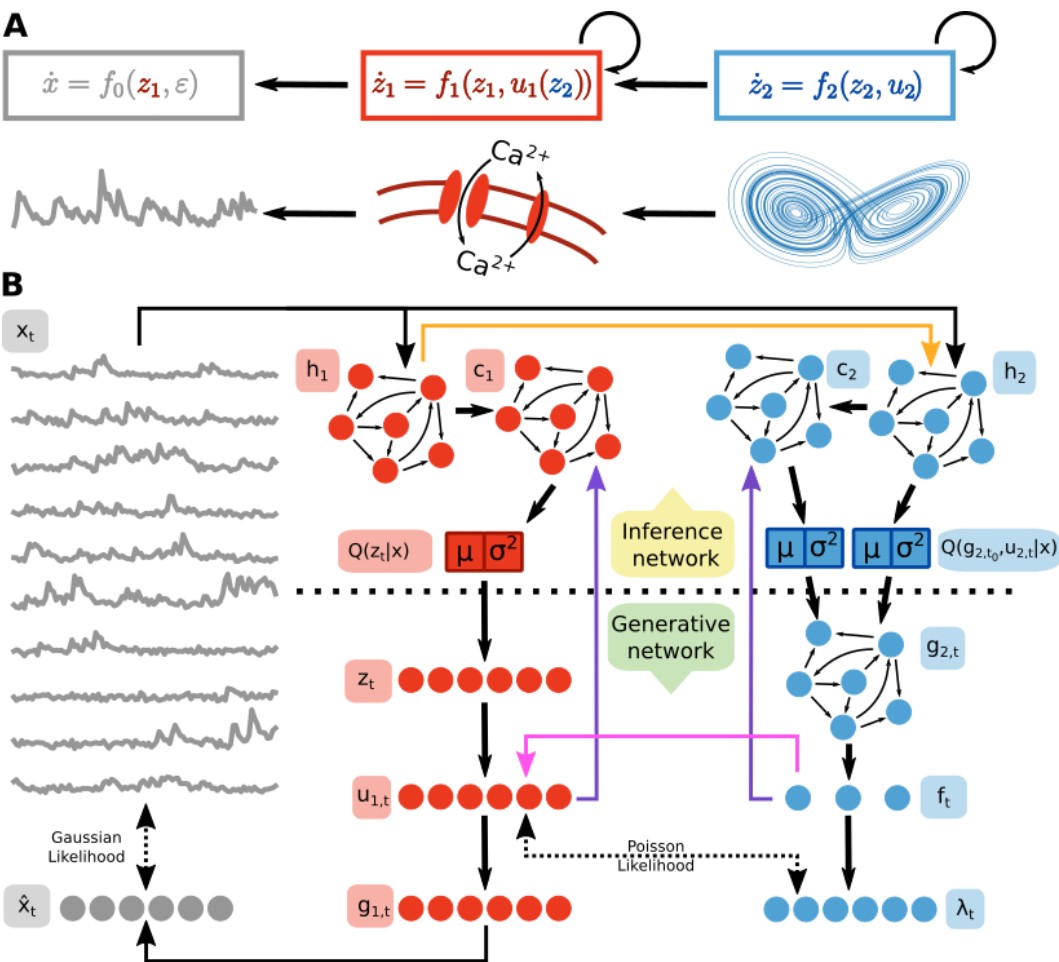

Figure 1: A) Hierarchy of dynamical systems (Top). schema of calcium and Lorenz dynamics (Bottom). B) Schema of our hierarchical model. Latent dynamics model in blue, calcium dynamics model in red.

## 2 MODEL

CaLFADS extends the variational ladder autoencoder (VLAE) approach (Zhao et al., 2017) with RNNs (Fig 1B), which enables inference of hierarchical latent dynamical systems[1]. The full directed acyclic graph for CaLFADS can be found in Figure S1. The VLAE approach to inferring hierarchies removes the sequential dependence of each latent variable, and instead loads parts of a flat latent code with more abstract features using more expressive neural networks (see appendix C for more background on VLAEs). For CaLFADS, this means we separate the shallower-level inference and the deeper-level inference into parallel pathways for inference and generation (1B, orange arrow), and subsequently, recombines these levels in the generative network (1B, pink arrow). This approach has two major advantages: 1) It solves the problem of latent variable collapse in which all latent features are loaded onto the lowest-order variables in the hierarchy. 2) It retains a very simple formulation for the variational/evidence lower bound (ELBO) that is easily reparameterisable (Sønderby et al., 2016; Zhao et al., 2017).

---

[1]Supporting code for this submission can be found at `https://anonymous.4open.science/repository/63e5b8d1-31b7-408a-99dc-a55980c2a1ca`

CaLFADS infers the dynamics and unknown events (e.g. spikes) that determine the calcium fluorescence signal (Fig 1B, red network nodes), and the underlying computational dynamical system that controls these events (Fig 1B, blue network nodes), through successive encoding and decoding layers in which latent variable distributions are modelled in separated 'rungs' of the ladder architecture. This VLAE architecture enables CaLFADS to disentangle the shallower-level calcium dynamics and the deeper-level computational dynamics. Algorithm 1 provides details on the forward pass of CaLFADS. Additionally, we provide a complete description of the model in Appendix D. Briefly, layers of Gated Recurrent Units (GRUs) create a deterministic hierarchical embedding of the data in the inference network (Fig 1B, above the dashed line). These embeddings are then concatenated with time-shifted, layer-specific deterministic projections of the latent state. This concatenated set of activities is then linearly transformed into the parameters of a factorised Gaussian posterior distribution for the latent variables in the generative network (Fig 1B, below the dashed line). This process provides the inference network with information about the history of the latent states via feedback connections (Fig 1B, purple feedback arrows).

As in any variational autoencoder (VAE) system, latent states are sampled from the posterior distributions determined by the inference network. For the inference of the computational dynamical system, these latent states represent the initial conditions ($g_{2,t_0}$) and unknown inputs ($u_{2,t}$) to a learned dynamical system modelled discretely by a GRU in the generative network ($g_{2,t}$). The hidden state of this GRU is then linearly transformed into a set of potentially low-dimensional time-varying latent factors ($ft$). Concurrently, the latent states for the calcium dynamics ($z_t$) are concatenated with the dynamic factors from the layer above ($f_t$), and undergo a linear transformation followed by a rectified-exponential non-linearity $f(x) = ReLU(\exp(x) - 1)$ to provide an approximation to spike counts ($u_{1,t}$). This non-linearity was chosen as a continuous relaxation to discrete calcium transient amplitudes inspired by the static distribution of amplitudes extracted via OASIS (Wei et al., 2019a). In our case, we model approximate spike counts using a log-normal distribution with truncation at 0. The approximate spike counts are then used as inputs to an order 1 autoregressive (AR(1)) process model of calcium dynamics ($g_{1,t}$). Since we found that the loss function was extremely sensitive to changes in the AR(1) model parameters, these parameters were fixed using values empirically derived from the OASIS-deconvolution algorithm (Friedrich et al., 2017). While it is possible to transform calcium dynamics into fluorescence using nonlinear models of fluorescence indicator kinetics (Speiser et al., 2017; Deneux et al., 2016), we found that using the AR(1) process to model the transformation of spike counts to fluorescence was sufficient for accurate reconstruction of the latent space, even in synthetic data with nonlinear calcium transient generation. Model hyperparameters for different datasets are shown in table S1.

## 2.1 LOSS FUNCTION AND TRAINING

One of the advantages of using VLAEs is that because the stochastic nodes operate in parallel, the ELBO formulation is the same as that for standard VAEs despite the hierarchical latent space (Zhao et al., 2017). As such, the CaLFADS cost function remains very similar to that of LFADS. It is described in Algorithm 2. One key addition to the LFADS loss function is that because the data is fluorescence data, which includes normal emissions noise, and the inferred shallower-level factors are calcium concentrations, the likelihood function of the data given the estimated calcium concentration, $P(x_t|\hat{x}_t)$, is modelled as a Gaussian distribution $x_t \sim \mathcal{N}(\hat{x}_t, \sigma_x^2)$, where $\sigma_x^2$ is determined by OASIS. Another key addition is that although spike counts $u_{1,t}$ are not modelled discretely, their probability distribution given the inferred spike rates, $P(u_{1,t}|\lambda_t)$, is modelled as an approximate Poisson process such that $\log(P(u_{1,t}|\lambda_t)) \approx u_{1,t}\log(\lambda_t) - (\lambda_t + u_{1,t})$. The approximation results from the use of an L1 norm on the spikes, instead of the log-gamma term that would normally be used in a Poisson process, which was necessary because most of the data sits within a range that is poorly handled by the log-gamma function. The use of this approximate Poisson process provides an additional regularization term in the cost function, which acts as an inductive bias to ensure that the inferred spike trains in the shallower-level dynamical system are constrained to follow the dynamics of the deeper-level dynamical system's spike rates $\lambda_t$. Per the VLAE framework, the shallower-level and deeper-level latent variables are treated as independent, with hierarchy imposed in the deterministic paths, i.e., $Q(g_{1,t}, g_{2,t=0}, u_{2,t}|x) = Q(g_{1,t}|x)Q(g_{2,t=0}|x)Q(u_{2,t}|x)$

Parameters of CaLFADS were optimised with ADAM, with an initial learning rate of 0.01. Learning rates were decreased by a factor of 0.95 when plateaus in training error were detected. As in LFADS training, KL and L2 terms in the cost function were 'warmed up', i.e., had a weight $w_{l \in (1,2)}$ between

0 and 1 applied that gradually increased (Bowman et al., 2016; Sønderby et al., 2016). Warm-up for the deeper parameters ($l = 2$, blue modules in Figure 1) was delayed until warm-up for shallower parameters ($l = 1$, red modules in Figure 1) was completed. We found in preliminary tests that this delayed warm-up sequence was necessary for good hierarchical inference compared to simultaneous warm-up. As with aggressive inference network training methods (He et al., 2019), this prevents issues with training of the deeper-level dynamical system parameters converging more slowly than the shallower-level dynamical system parameters in the initial stages of training.

## 3 RESULTS

### 3.1 LORENZ ATTRACTOR SYNTHETIC DATA

#### 3.1.1 DATA DESCRIPTION

As an initial test of CaLFADS, we examined its ability to infer the hierarchical dynamical systems of synthetic calcium fluorescence data generated from a simple model of spiking neurons. We examined synthetic data generated from a network with a Lorenz attractor embedded in its dynamics. This tests our ability to recover a ground-truth deeper-level dynamical system from the data, and has been used as a benchmark by many others (Zhao & Park, 2017; Pandarinath et al., 2018; Clark et al., 2019; She & Wu, 2019; Koppe et al., 2019).

In this system, network dynamics were used to generate a set of spike rates in simulated neurons. We then used a model of calcium dynamics and emissions noise to transform the spikes into synthetic fluorescence data. We tested CaLFADS on data generated from two models of synthetic calcium dynamics: a simple linear model equivalent to an AR(1) process, and a nonlinear model in which an AR(1) process is transformed by a Hill function nonlinearity. We then added white noise to the resulting traces. This procedure is described in more detail in Appendix H.

#### 3.1.2 MODEL COMPARISON

We measured the performance of CaLFADS by computing $R^2$ goodness-of-fit with the embedded Lorenz attractor state variables. We also measured the ability of CaLFADS to reconstruct other ground-truth variables in the synthetic data (spike counts, spike rates, and fluorescence traces) using $R^2$, but these were not as critical for assessing model performance.

CaLFADS performance was compared against two baselines. First, we use LFADS with a Gaussian likelihood observation model to account for fluorescence (Gaussian-LFADS). Second, we consider the situation where spike counts are first estimated separately using the OASIS deconvolution algorithm (Friedrich et al., 2017), a robust, popular, and computationally inexpensive method of spike extraction widely used by systems neuroscientists (Pachitariu et al., 2018; Evans et al., 2019). LFADS is then used to infer the deeper-level dynamical system and reconstruct the spike rates from the deconvolved spike trains. We refer to this approach as OASIS+LFADS throughout.

Table 1 compares $R^2$ goodness-of-fit in reconstructing ground-truth dynamic latent variables in held-out validation data. Table 1 shows CaLFADS is able to reconstruct the Lorenz attractor state for synthetic data with either linear or nonlinear synthetic calcium models (see Table 1 caption for significance test results). Indeed, the inclusion of nonlinearities in calcium dynamics does not affect CaLFADS ability to reconstruct the Lorenz attractor state as much as it does with OASIS+LFADS. CaLFADS is also better able to reconstruct other network state variables compared to OASIS+LFADS as can also be seen in Table 1. However, it is worth noting that high accuracy in reconstructing other network state variables does not appear to be necessary for accurate reconstruction of the underlying Lorenz attractor. This indicates that CaLFADS is better suited to handling uncertainty due to spike-timing and is capable of separating sources of slow dynamics to reconstruct the embedded latent space. Example outputs of CaLFADS and OASIS+LFADS are shown in Appendix I.1 and I.2.

Note that CaLFADS reconstructs the Lorenz dynamics almost as well as LFADS does when applied to the spiking data. Of course, it is to be expected that LFADS applied to true spiking data performs better than CaLFADS, since there is an additional source of observation noise from the generation of fluorescence transients. But, the fact we can get very close to the same level of perfor-

Table 1: Comparison of model performance ($R^2$ mean+sem) on synthetic Lorenz datasets generated with 15 different seeds. A hyphen indicates that the variable cannot be compared, as the model does not infer it. The top row is italicised as the performance of LFADS on this task is considered the upper limit, having no additional observation noise from fluorescence. Results of paired t-tests for testing significance of differences in performance between OASIS+LFADS and CaLFADS within seeds for a) linear calcium model: Lorenz state - $t_{14} = 2.85$, $p = 0.013$; Spike Counts - $t_{14} = 20.67$, $p < 0.001$; Rates - $t_{14} = 4.81$, $p < 0.001$; Fluorescence - $t_{14} = 16.2$, $p < 0.001$. b) nonlinear calcium model: Lorenz state - $t_{14} = 3.29$, $p = 0.005$; Spike Counts - $t_{14} = 17.16$, $p < 0.001$; Rates - $t_{14} = 6.35$, $p < 0.001$; Fluorescence - $t_{14} = 11.16$, $p < 0.001$.

| Calcium | Model | Lorenz state | Spike Counts | Firing Rates | Fluorescence |
|---|---|---|---|---|---|
| NA | *LFADS* | *0.975 ± 0.001* | - | *0.956 ± 0.003* | - |
| Linear | Gaussian-LFADS | 0.818 ± 0.005 | - | 0.701 ± 0.004 | - |
| | OASIS+LFADS | 0.960 ± 0.001 | 0.580 ± 0.006 | 0.912 ± 0.002 | 0.807 ± 0.005 |
| | CaLFADS | **0.965 ± 0.001** | **0.817 ± 0.007** | **0.937 ± 0.002** | **0.909 ± 0.003** |
| Nonlinear | Gaussian-LFADS | 0.808 ± 0.005 | - | 0.650 ± 0.006 | - |
| | OASIS+LFADS | 0.877 ± 0.003 | 0.351 ± 0.004 | 0.682 ± 0.005 | 0.843 ± 0.004 |
| | CaLFADS | **0.902 ± 0.006** | **0.561 ± 0.010** | **0.756 ± 0.008** | **0.932 ± 0.005** |

mance indicates that CaLFADS is effective at performing the same inference on calcium data that LFADS performs on spiking data. We speculate that the uncertainty over AR1 process parameters is overcome in CaLFADS by constraining the reconstructed spike counts with the latent dynamics to make the inference of the observed dynamics process "population-aware", something which cannot be done when using LFADS applied to spike counts obtained by OASIS deconvolution. This may mitigate errors in spike count reconstruction that occur by deconvolution due to the absence of information about whole-network dynamics. Thus, CaLFADS can use population-level dynamics when conducting the inference of network variables, and this helps it to separate out the shallower-level dynamics more accurately.

## 3.2 ROTATIONAL DYNAMICS IN MONKEY MOTOR CORTEX

Next, we wanted to test CaLFADS on a real neural dataset to which LFADS has previously been applied, and successfully used to uncover meaningful latent dynamics. Specifically, it has been shown previously that rotational dynamics underlie neuronal responses in monkey and human motor cortex during reaching behaviour (Churchland et al., 2012). LFADS has been successful in uncovering these known rotational dynamics for single-trial spikes recorded from primary motor cortex (M1) and dorsal premotor cortex (PMd) in macaques (Pandarinath et al., 2018). To test whether CaLFADS could do the same, we took the original spiking data and converted it into semi-synthetic calcium traces, using the same model of calcium dynamics that was used for synthetic data generation in the previous sections (Fig 2A). We used data from the monkey electrophysiology dataset previously described, along with the reaching task, by Churchland et al. (2012), which was kindly provided to us by the original authors. Briefly, monkeys were trained to reach a target under 108 different reach conditions while multi-electrode recordings were made in M1 and PMd. Reaches started from a specified location on a screen, and monkeys were rewarded for correctly reaching toward a target while avoiding on-screen obstacles (Fig 2B).

First, to replicate the previous LFADS results, we applied jPCA (Churchland et al., 2012) to condition-averaged firing rates (trial averages for each of the 108 reach conditions), as well as single-trial firing rates inferred from spike data using LFADS. jPCA is a dimensionality reduction technique that finds orthogonal projections capturing rotational dynamics that explain variability in firing rates. The original LFADS paper showed that rotational dynamics explained a large amount of the variance in firing rates. As in the original papers, we identified both condition-averaged (Fig 2C – top) and single-trial (Fig 2C – bottom) rotational dynamics from firing rates inferred by LFADS, which explained a large amount of the variance ($R^2 = 0.81$), thereby replicating the original findings. Then, we evaluated our CaLFADS model in uncovering the rotational dynamics of both condition-averaged and single trials from firing rates. As shown in Fig 2, our model also successfully uncovers rotational dynamics from calcium traces, explaining a large amount of variance ($R^2 = 0.78$) for both

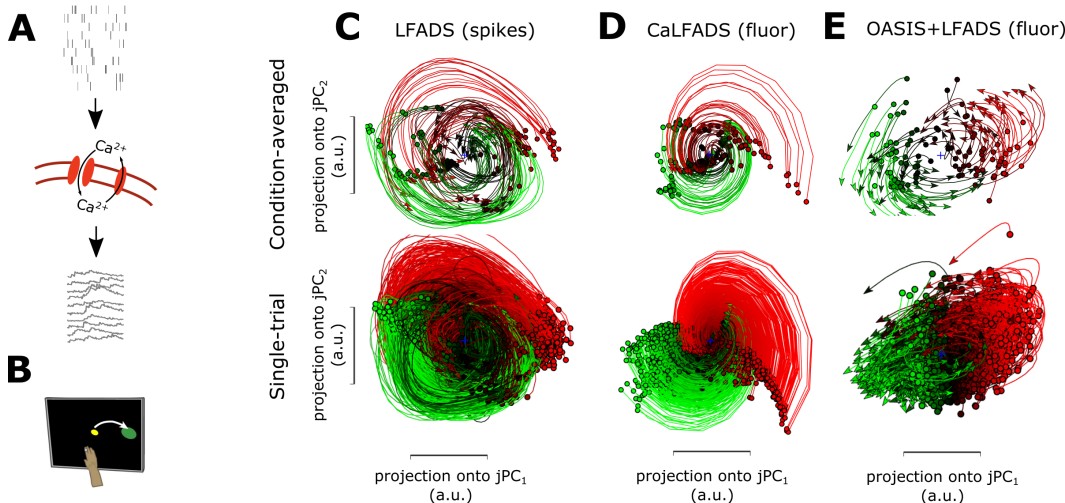

Figure 2: A) Schematic for the process of converting spikes to calcium traces, and B) for the reaching task. C) Rotational dynamics inferred from condition-averaged (Top) and single-trial spikes (Bottom) using LFADS. D) Rotational dynamics inferred from condition-averaged (Top) and single-trial calcium traces (Bottom) using CaLFADS. E) Rotational dynamics inferred from condition-averaged (Top) and single-trial calcium traces (Bottom) using OASIS+LFADS. Traces are coloured based on their initial state values along jPC1 (from green to red for increasingly larger values)

condition-averaged (Fig 2D – top) and single trials (Fig 2D – bottom). These results demonstrate that CaLFADS is capable of identifying latent dynamics in real neural data, similar to LFADS, even when the spiking data is transformed by calcium dynamics and emissions noise. We also found that the OASIS+LFADS approach was not as successful as CaLFADS in uncovering rotational dynamics, explaining only half the variance ($R^2 = 0.53$, Fig 2E). This result corroborates the importance of the hierarchical modelling of calcium and neuronal dynamics in CaLFADS.

### 3.3 UNEXPECTED STIMULUS DETECTION IN MOUSE PRIMARY VISUAL CORTEX

Finally, we wanted to test CaLFADS on an entirely new calcium imaging dataset, to determine whether CaLFADS can infer dynamic computational factors that carry information about relevant features of the outside world. We chose to analyse data from mouse primary visual cortex (VisP), a region widely studied in systems neuroscience research where calcium imaging is a standard tool. There is evidence that the visual cortex of mammals performs a predictive function, anticipating expected stimuli (Dayan et al., 1995; Rao & Ballard, 1999; Fiser et al., 2016; Leinweber et al., 2017). As a result, unexpected stimuli can induce perturbations in network dynamics (Fiser et al., 2016; Leinweber et al., 2017). Thus, we wanted to determine whether CaLFADS could infer dynamic computational factors that carry information distinguishing unexpected from expected visual stimuli.

To this end, we trained CaLFADS on calcium imaging data from the mouse visual cortex collected by the Allen Institute for Brain Science (for a detailed description, see Appendix J.1). While awake behaving mice were presented with visual stimuli on a screen (Fig 3A), calcium fluorescence responses in cortical layer 2/3 of VisP were recorded using 2-photon microscopy (Fig 3B). The mice were familiarised with sequences of stimulus frames that followed simple probabilistic rules. Briefly, each sequence consisted of four randomised Gabor pattern frames followed by a grey screen (A, B, C, D, grey) with orientations drawn for each trial from the same distribution. After familiarization with this sequence, the expected D frame was replaced in 10% of trials by a Gabor pattern (E) with orthogonal orientations (Fig 3C). CaLFADS inferred smooth latent factors that did not show signs of over-fitting to noise (Appendix J.2, Fig S7). To determine if salient information about these unexpected stimulus features was present in the inferred factors from CaLFADS, we trained a non-linear decoder to classify whether the inferred factors came from a trial with an expected D frame or an unexpected E frame (for details, see Appendix J.3). We also trained the non-linear decoder on factors inferred by OASIS+LFADS, and on fluorescence traces reduced in dimensionality with prin-

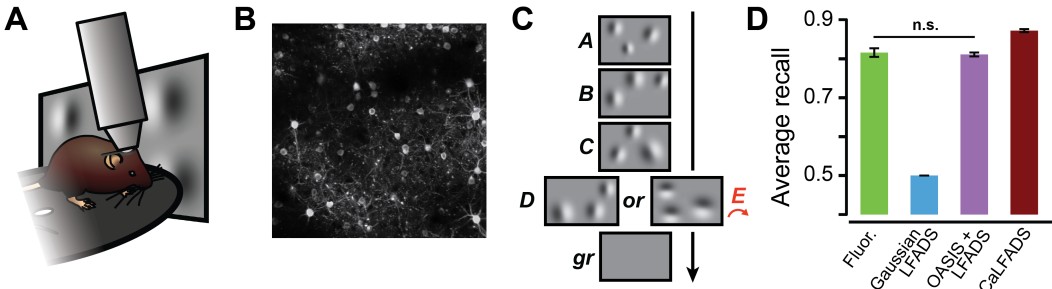

Figure 3: A) Schematic of 2-photon calcium imaging recording setup. Mice are head-fixed on a running wheel under the imaging objective, and visual stimuli are projected onto a screen to the side, B) 2-photon calcium imaging recording plane, C) Schematic example of visual stimuli for expected (A-B-C-D-grey) and unexpected (A-B-C-E-grey) trials. D) Average recall (mean ± sem) of expected vs. unexpected trials across non-linear decoders trained on principal components of fluorescence traces ($0.815 \pm 0.011$, green), Gaussian-LFADS factors ($0.500 \pm 0.000$, blue), OASIS+LFADS factors ($0.810 \pm 0.005$, purple), and CaLFADS factors ($0.871 \pm 0.004$, maroon). All pairs but one (fluor. vs. OASIS+LFADS factors, marked with n.s.) were significantly different at $p < 0.05$ in 2-tailed independent t-tests with Bonferroni correction for multiple comparisons.

cipal components analysis. To compare the models fairly, despite the imbalance between expected and unexpected trial frequencies, we measured average recall. We find that using the CaLFADS factors leads to the highest performance on stimulus-trial identity decoding, with the average recall being significantly higher than when OASIS+LFADS factors are used (Fig 3D). This indicates that CaLFADS infers latent dynamics from real visual cortex calcium imaging data that reflect the predictability of visual stimuli, corroborating the ability of CaLFADS to infer meaningful dynamic factors in real data.

## 4 DISCUSSION

In this paper we presented CaLFADS, a hierarchical recurrent variational autoencoder model capable of reconstructing latent computational dynamics. We confirmed CaLFADS' ability to reconstruct known underlying dynamics using synthetic datasets where ground-truth was known. We also showed that CaLFADS is able to infer sensorimotor dynamics from real neural recordings. This indicates that CaLFADS is a promising method for analyzing calcium imaging data in neuroscience.

There are two key advantages of CaLFADS over the use of deconvolution of calcium traces followed by application of LFADS. The first is that we can obtain measures of the uncertainty in both the latent dynamics, and the latent spike counts. The second is that CaLFADS performs better than OASIS+LFADS when nonlinearities in calcium dynamics are present, as is the case in real experimental data where there are many sources of calcium influx related to spikes.

We designed CaLFADS to fit into a much broader class of artificial neural network models, namely *sequential variational ladder autoencoders*, in which different layers of recurrent neural networks can be used to infer and generate different layers of a hierarchical dynamical system. In this sense, this class of models is modular and composable, meaning it could be readily adapted to other domains. For example, it should be possible to replace LFADS as the deeper-level dynamical system model with any other differentiable model. Likewise the same is true for our AR1 based model of calcium dynamics: we could replace the calcium dynamics model with other models as required by the experimental set-up. Furthermore, there is no need to stop at two layers in the hierarchy; the brain is comprised of many interconnected recurrent neural networks sending long range signals to one another, and it should be possible to add in additional modules to capture additional hierarchies and dynamics within the brain.

In summary, CaLFADS is a new, open source tool for inferring hierarchical dynamics from calcium imaging data that also has great potential for being modified and applied to other data modalities. This could be of real benefit to the thousands of neuroscience laboratories around the world conducting calcium imaging experiments.

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

# A  VARIABLE GLOSSARY

Table S1: Glossary of variables, with descriptions and dimensionality across datasets

| Variable | Description | Dimensions | | | |
|---|---|---|---|---|---|
| | | Lorenz | RNN | M1/PMd | V1 |
| $x_t$ | Fluorescence signal | 30 | 50 | 202 | 96 |
| $h_{1,t}$ | Calcium dynamics embedding encoder state | 128 | 64 | 200 | 128 |
| $h_{2,t}$ | Computational dynamics embedding encoder state | 64 | 128 | 100 | 128 |
| $c_{1,t}$ | Calcium dynamics embedding controller state | 128 | 64 | 200 | 128 |
| $c_{2,t}$ | Computational dynamics embedding controller state | 0 | 128 | 0 | 64 |
| $g_{1,t}$ | Calcium dynamics embedding state | 100 | 64 | 200 | 128 |
| $z_{2,t}$ | Computational dynamics embedding state | 64 | 200 | 100 | 200 |
| $u_{1,t}$ | Approximate spike counts | 30 | 50 | 202 | 96 |
| $u_{2,t}$ | Computational dynamics perturbations | 0 | 1 | 0 | 1 |
| $g_{1,t}$ | Calcium dynamics | 30 | 50 | 202 | 96 |
| $f_t$ | Dynamic factors | 3 | 20 | 40 | 32 |
| $\hat{x}_t$ | Reconstructed fluorescence signal | 30 | 50 | 202 | 96 |

## B  RELATED WORK AND CONTRIBUTION

This work brings together two important strands of computational neuroscience research that have been of interest to the machine learning community, namely probabilistic latent factor models of neural dynamics and inference of spikes in calcium fluorescence traces. We solve these two problems as a hierarchical inference problem using variational ladder autoencoders. We give a brief introduction to variational ladder autoencoders in Appendix C.

LFADS is a power latent factor model of neural dynamics proposed by Pandarinath et al. (2018). It is an extension of the sequential variational autoencoder DRAW (Gregor et al., 2015) applied to spiking data that adds an additional RNN to the encoder architecture to infer unknown initial conditions to a generator RNN. This enables interpretation of the generative model as a nonlinear dynamical system that generates observed spikes. Low dimensional projections of the generator hidden state (latent factors) were found to be capable of decoding single-trial reaching movements with very high accuracy. This was a hugely influential and important contribution to the field of neuroscience as it demonstrated a means to find latent dynamical systems that solved computational tasks required for executing behaviour. However, LFADS could not be applied to neural data where the spiking was itself unknown, as in two-photon calcium imaging.

DeepSpike by Speiser et al. (2017) is a variational autoencoder adapted from MLSpike, a probabilistic model of calcium fluorescence recordings (Deneux et al., 2016). It was a contribution to the SpikeFinder competition (Berens et al., 2018) in which participants were given the task of inferring an unknown spike train from calcium fluorescence recordings. Unlike in most two-photon recordings, these data also included ground-truth whole-cell electrophysiology, i.e., ground-truth spike trains. While other techniques had higher performance, it was the first attempt at solving this problem using variational inference. One issue with this model is that it used a Bernoulli distribution to model the latent spike train. Since this is not reparameterisable, Speiser et al. (2017) needed to use a high variance Monte Carlo objective to train the model. The use of this objective means it would not scale well to 2-photon measurements from many cells. Furthermore, such a model does not capture how neurons can spike multiple times within the time-bin of a typical two-photon imaging experiment.

We note some similarities between our generative model and Aitchison et al. (2017). A key difference here is that we try to find nonlinear latent dynamics, whereas their model comprises a linear dynamical system in the latent space. Furthermore, theirs is a fully Bayesian approach, whereas CaLFADS has many more deterministic components.

We also note the capabilities of VIND (Hernandez et al., 2020) in inferring non-linear latent dynamics in wide-field calcium imaging data, which has insufficient spatial resolution to observe individual cells. Hernandez et al. (2020) propose a variational inference technique in which latent dynamics are modelled by a locally linear dynamical system and find informative latent trajectories in the recorded smoothed population activity. We note however that this approach is not used to infer latent dynamics in two-photon recordings, where it would be necessary to disentangle calcium dynamics from population dynamics.

Finally, the variational autoencoder LeMoNADe proposed by Kirschbaum et al. (2019) similarly proposes an end-to-end system of inferring latent activities from calcium imaging. In contrast to CaLFADS, they are attempting to infer the presence of neuronal ensembles, sets of cells grouped together by their activities. While this is a related goal to inferring a latent dynamical system, there is no requirement for the latent factors to be as interpretable as in LeMoNADe. Furthermore, the neuronal ensembles identified by LeMoNADe were not demonstrated to help solve relevant behavioural tasks. However, LeMoNADe were the first to demonstrate how neuronal activities could be modelled by a continuous relaxation to the Bernoulli distribution using the Gumbel-Softmax trick.

CaLFADS ties together this related work. We propose a hierarchical variational autoencoder capable of inferring latent dynamic factors in calcium imaging data by separating shallower-level calcium dynamics and deeper-level computational dynamics. We propose a novel continuous relaxation to spike counts that allows an arbitrary number of spikes to be counted within the typically low sampling rate of calcium fluorescence recordings (10-30 Hz). We also propose a novel KL annealing technique for allowing hierarchical latent dynamics to be inferred.

## C    INTRODUCTION TO VARIATIONAL LADDER AUTOENCODERS

Variational Ladder Autoencoders were proposed as a way to ensure that higher- and lower-level features could be stored in different parts of the latent state. They propose a ladder architecture in which more expressive networks sit deeper in the hierarchy and no hierarchy is assumed among latent variables directly. For layers $1 : L$ in the hierarchy

$$P(\mathbf{z_1}, ..., \mathbf{z_L}) = \prod_l P(\mathbf{z_l}) \tag{1}$$

$$Q(\mathbf{z_l}|\mathbf{x}) = f_l(\mathbf{h_l}) \tag{2}$$

$$\mathbf{h_l} = g_l(\mathbf{h_{l-1}}) \tag{3}$$

where $\mathbf{h_0} = \mathbf{x}$, and $g_l$ and $f_l$ are neural networks with increasing degrees of expressivity as $l$ increases. This encourages more abstract features to be represented in certain portions of the latent state.

This is in contrast to traditional hierarchical autoencoders that assume a Markovian structure in the latent variables

$$P(\mathbf{x}, \mathbf{z}) = P(\mathbf{x}|\mathbf{z}_1) \prod_{l=1}^{L-1} P(\mathbf{z}_l|\mathbf{z}_{l+1}) P(\mathbf{z}_L) \tag{4}$$

The key insight from (Zhao et al., 2017) is that under idealised assumptions for sampling and optimization of the ELBO, the ELBO for hierarchical VAEs is optimised by minimizing only $D_{\mathrm{KL}}(Q(\mathbf{z_1}|\mathbf{x})||P(\mathbf{z_1}|\mathbf{x}))$. This means that traditional hierarchical VAEs will load all representations onto the lowest level of the hierarchy.

In contrast, by splitting the latent code by abstractness and parameterising the factorised posterior distribution with increasingly expressive functions, the VLAE can be trained with the simple ELBO formulation used in non-hierarchical VAEs.

# D  CaLFADS PROBABILISTIC MODEL

Here we outline the assumed probabilistic model for CaLFADS. In doing so, we will briefly review the model of LFADS (Pandarinath et al., 2018) and the extensions that make it possible to infer dynamic latent variables from calcium fluorescence traces.

## D.1  LFADS GENERATIVE MODEL

LFADS assumes that neural activity $\mathbf{x}$ is generated from a low dimensional non-linear dynamical system with unknown initial conditions subjected to unknown control inputs. The state of the low-dimensional non-linear dynamical system is modelled by the hidden state $\mathbf{g_t}$ of a Gated Recurrent Unit receiving control inputs $u_t$. The prior for the initial conditions $\mathbf{g_0}$ is assumed to be normal with mean $\mu_{g_0}$ and variance $\sigma_{g_0}^2$. External inputs are assumed to have an autoregressive process prior with process mean $\mu_u$, process variance $\sigma_u^2$, and time constant $\tau_x$,

$$P(\mathbf{g_t}) = \mathcal{N}(\mu_{g_0}, \sigma_{g_0}^2) \tag{5}$$

$$P(\mathbf{u_1}) = \mathcal{N}(\mu_x, \sigma_u^2) \tag{6}$$

$$P(\mathbf{u_t}|\mathbf{u_{t-1}}) = \mathcal{N}(\mu_u + e^{\frac{-1}{\tau_u}}(\mathbf{u_{t-1}} - \mu_u),\ \sigma_u^2(1 - e^{\frac{-1}{\tau_u}})) \tag{7}$$

$$\mathbf{g_t} = GRU(\mathbf{g_{t-1}}, \mathbf{u_t}) \tag{8}$$

The hidden states of the GRU are then projected to a low-dimensional set of factors $f_t$

$$\mathbf{f_t} = \mathbf{W}^f \mathbf{g_t} + \mathbf{b}^f \tag{9}$$

In Pandarinath et al. (2018), the neural activity being modelled was single-trial spikes extracted from electrophysiological data in macaque motor cortex. LFADS assumes a simple observation model for spiking data where the spike count observed in a given time-bin $x_t$ follows an independent Poisson process parameterised by the instantaneous firing rate $\lambda_t$. To obtain the instantaneous firing rate, low-dimensional factors $f_t$ are linearly projected to the dimensionality of the data and a non-linearity enforcing positivity is applied, in this case the exponential function

$$\lambda_\mathbf{t} = \exp(\mathbf{W}^\lambda \mathbf{f_t} + \mathbf{b}^\lambda) \tag{10}$$

$$P(\mathbf{x_t}|\lambda_\mathbf{t}) = Poisson(\lambda_\mathbf{t}) \tag{11}$$

## D.2  CALCIUM FLUORESCENCE GENERATIVE MODEL

Before describing how the low dimensional dynamic factors are used to generate calcium fluorescence data, we first describe our probabilistic model for calcium fluorescence transients. We assume that calcium fluorescence transients are generated by an autoregressive process $c_t$ occluded by white emissions noise $\epsilon$ and perturbed by an unknown spike train $\mathbf{s_t}$.

Probabilistic models of unknown spike counts in calcium fluorescence data typically assume a Bernoulli distribution, or some continuous approximation thereof (Speiser et al., 2017). This is somewhat problematic given that fluorescence is sampled at a rate of 10-30 Hz, leaving time-bins large enough to observe more than one spike per time-bin. Recently, Wei et al. (2019a) demonstrated that spike counts inferred by the deconvolution algorithm OASIS (Friedrich et al., 2017) could be closely modelled by a zero-inflated gamma model. Similar to this, we modelled spike counts using a similar continuous approximation

$$P(\mathbf{s_t}) = ReLU(LogNormal(\mu_{s_t}, \sigma_{s_t}^2) - 1) \tag{12}$$

This approximation is easily reparameterisable since we can sample from a lognormal distribution by transforming samples from a normal distribution with an exponential function. The ReLU function and subtraction of 1 from samples ensures that many counts are zero. In contrast, since the zero-inflated gamma model which is a mixture distribution, it would require modelling a hierarchical discrete-continuous latent variable which is not as simply reparameterisable for backpropagation.

Spike counts are then used to perturb an autoregressive process modelling the slow decay of calcium fluorescence after a spike. A simple autoregressive model for calcium fluorescence is the AR(1) process. With discretised time, this can be modelled with Euler updates

$$\mathbf{c_t} = \mathbf{c_{t-1}}(1 - \beta) + \alpha \mathbf{s_t} \tag{13}$$

where $\beta = \frac{\Delta t}{\tau_c}$, with $\Delta t$ the size of time bin and $\tau_c$ the decay of calcium fluorescence, and with $\Delta t < \tau_c$. $\alpha$ is a constant gain term controlling the effect of spikes on fluorescence.

A nonlinear function modelling dye kinetics $d(\cdot)$ can then be applied. For these experiments we have not done so, but it would be a relatively simple extension. As such, the observation model for fluorescence $F$ becomes

$$P(\mathbf{F_t}|\mathbf{c_t}) = \mathcal{N}(d(\mathbf{c_t}), \sigma_F^2) \tag{14}$$

where in our case, $d(\mathbf{c_t}) = \mathbf{c_t}$.

### D.3 CALFADS GENERATIVE MODEL

To tie LFADS and our observation model for calcium fluorescence together, we will adjust the notation slightly to make the hierarchical positioning of latent variables clearer. The variables of the deeper-level model of LFADS are denoted by the subscript $l = 2$, whereas the variables of the shallower-level model of observed calcium fluorescence are denoted by the subscript $l = 1$. For consistency and notation simplicity, we denote variables that are the state of a dynamical system by $g_{l,t}$ and inputs to the dynamical system as $u_{l,t}$.

We observe neural activity $\mathbf{x}$ in the form of single-trial calcium fluorescence traces. We assume that fluorescence follows a simple autoregressive process $\mathbf{g_{l=1}}$ perturbed by an unknown set of spike counts $\mathbf{u_{l=1}}$ and subject to additive white emissions noise. We also assume that spike counts $\mathbf{u_{l=1}}$ are influenced by the state of a nonlinear dynamical system with unknown initial conditions $\mathbf{g_{l=2,t=0}}$ and unknown inputs $\mathbf{u_{l=2}}$.

As in LFADS, the nonlinear dynamical system is modelled by a GRU. The hidden state of the GRU $g_{l=2,t}$ is projected to a low-dimensional set of factors $\mathbf{f_t}$

$$P(\mathbf{g_{2,0}}) = \mathcal{N}(\mu_{g_{2,0}}, \sigma_{g_{2,0}}^2) \tag{15}$$

$$P(\mathbf{u_{2,1}}) = \mathcal{N}(\mu_{u_2}, \sigma_{u_2}^2) \tag{16}$$

$$P(\mathbf{u_{2,t}}|\mathbf{u_{2,t-1}}) = \mathcal{N}(\mu_{u_2} + e^{\frac{-1}{\tau_{u_2}}}(\mathbf{u_{2,t-1}} - \mu_{u_2}), \ \sigma_{u_2}^2(1 - e^{\frac{-1}{\tau_{u_2}}})) \tag{17}$$

$$\mathbf{g_{2,t}} = GRU(\mathbf{g_{l,t-1}}, \mathbf{u_{2,t}}) \tag{18}$$

$$\mathbf{f_t} = \mathbf{W}^f \mathbf{g_{2,t}} + \mathbf{b}^f \tag{19}$$

For spike counts to be informed by the dynamic factors $\mathbf{f_t}$, we sample spike counts according to

$$\mathbf{z_t} \sim P(\mathbf{z_t}) = \mathcal{N}(\mu_z, \sigma_z^2) \tag{20}$$

$$\mathbf{u_{1,t}} = ReLU(\exp(\mathbf{W}^s[\mathbf{z_t}, \mathbf{f_t}] + \mathbf{b}^s) - 1) \tag{21}$$

using the continuous approximation to spike counts described in section D.2.

Although this factorization separates the generation of approximate spikes counts from the dynamic latent factors, for the formulation of the loss function we treat $\mathbf{u_{1,t}}$ as if it is generated from a homogeneous Poisson process with intensity $\lambda_t = \exp(\mathbf{W}^\lambda \mathbf{f_t} + \mathbf{b}^\lambda)$, i.e.,

$$P(\mathbf{u_{1,t}}|\lambda_t) \approx Poisson(\lambda_t) \tag{22}$$

We then use these approximated spikes counts to perturb the state $g_{l=1,t}$ of an AR(1) process

$$\mathbf{g_{1,t}} = \mathbf{g_{1,t-1}}(1 - \beta) + \alpha \mathbf{u_{1,t}} \tag{23}$$

The likelihood of the observed fluorescence in a given time bin $\mathbf{x_t}$ is then modelled as normal random variable with a mean dependent on the state of the AR(1) process.

$$P(\mathbf{x_t}|\mathbf{g_{1,t}}) = \mathcal{N}(d(\mathbf{g_{1,t}}), \sigma_x^2) \tag{24}$$

where again $d(\cdot)$ is an optional function modelling dye kinetics. In our case $d(\mathbf{g_{1,t}}) = \mathbf{g_{1,t}}$.

### D.4 CALFADS INFERENCE NETWORK

The inference network of CaLFADS parameterises the approximate posterior distributions $Q(g_{2,0}|x)$, $Q(u_{2,t}|x)$ and $Q(z|x)$. As per the ladder architecture (Zhao et al., 2017) these posterior distributions take the form

$$Q(\mathbf{z}|\mathbf{x}) = q_\theta(p_\phi(\mathbf{x})) \tag{25}$$

$$Q(\mathbf{u}_{2,0}|\mathbf{x}) = r_\psi(p_\phi(\mathbf{x})) \tag{26}$$

where $p(\cdot)$, $q(\cdot)$, $r(\cdot)$ are neural networks parameterised by $\phi$, $\theta$, $\psi$ respectively. The combination of latent space factorization and parameter sharing ensures that dynamics can be disentangled effectively.

As in LFADS, the approximate posterior distribution over the initial conditions to the nonlinear dynamical system $Q(g_{2,0})$ is parameterised separately using a bidirectional Gated Recurrent Unit

$$Q(\mathbf{g}_{2,0}|\mathbf{x}) = \mathcal{N}(\mu_{g_{2,0}}, \sigma^2_{g_{2,0}}) \tag{27}$$

$$\mu_{g_{2,0}} = \mathbf{W}^{\mu_{g_{2,0}}} \mathbf{h}^{g_{2,0}} + \mathbf{b}^{\mu_{g_{2,0}}} \tag{28}$$

$$\sigma_{g_{2,0}} = \exp(\frac{1}{2} \mathbf{W}^{\sigma_{g_{2,0}}} \mathbf{h}^{g_{2,0}} + \mathbf{b}^{\sigma_{g_{2,0}}}) \tag{29}$$

where $h^{g_{2,0}}$ is the final hidden state of a bidirectional Gated Recurrent unit

$$\mathbf{h}^{g_{2,0}} = [\mathbf{h}^{g_{2,0},fwd}_T, \mathbf{h}^{g_{2,0},bwd}_0] \tag{30}$$

$$\mathbf{h}^{g_{2,0}}_t = BiGRU(\mathbf{h}^{g_{2,0}}_{t-1}, \mathbf{x}_t) \tag{31}$$

$$= GRU([\mathbf{h}^{g_{2,0},fwd}_{t-1}, \mathbf{h}^{g_{2,0},bwd}_{t+1}], \mathbf{x}_t) \tag{32}$$

where $\mathbf{h}^{g_{2,0},fwd}_t$ is the state of a GRU running forward sequentially over the input, and $\mathbf{h}^{g_{2,0},bwd}_t$ is the state of a GRU running backward sequentially over the data.

The functions $f_\phi(\cdot)$, $g_\theta(\cdot)$, $h_\psi(\cdot)$ are modelled by pairs of recurrent neural networks. As in LFADS, we use bidirectional GRUs followed by unidirectional controller GRUs receiving representations from the generative network. The bidirectional GRU is defined as before

$$\mathbf{h}^u_{1,t} = BiGRU(\mathbf{h}^u_{1,t-1}, \mathbf{x}_t) \tag{33}$$

$$\mathbf{h}^u_{2,t} = BiGRU(\mathbf{h}^u_{2,t-1}, \mathbf{h}_{1,t}) \tag{34}$$

From this point the inference model splits into the two levels of the hierarchy. The hidden state $\mathbf{h}^u_t$ is passed through the a controller GRU which takes feedback from the generated samples

$$\mathbf{c}_{1,t} = GRU(\mathbf{c}_{1,t-1}, [\mathbf{h}^u_{1,t}, \mathbf{u}_{t-1}]) \tag{35}$$

$$\mathbf{c}_{2,t} = GRU(\mathbf{c}_{2,t-1}, [\mathbf{h}^u_{2,t}, \mathbf{f}_{t-1}]) \tag{36}$$

The states of the controller GRU are then linearly transformed onto the parameters of a normal distribution

$$Q(\mathbf{z}_t|\mathbf{x}) = \mathcal{N}(\mu_{z_t}, \sigma^2_{z_t}) \tag{37}$$

$$Q(\mathbf{u}_{2,t}|\mathbf{x}) = \mathcal{N}(\mu_{2,t}, \sigma^2_{2,t}) \tag{38}$$

$$\mu_{z_t} = \mathbf{W}^{\mu_z} \mathbf{c}_{1,t} + \mathbf{b}^{\mu_z} \tag{39}$$

$$\sigma_{z_t} = \exp(\frac{1}{2} \mathbf{W}^{\sigma_z} \mathbf{c}_{1,t} + \mathbf{b}^{\sigma_z}) \tag{40}$$

$$\mu_{u_{2,t}} = \mathbf{W}^{\mu_{u_2}} \mathbf{c}_{2,t} + \mathbf{b}^{\mu_{u_2}} \tag{41}$$

$$\sigma_{u_{2,t}} = \exp(\frac{1}{2} \mathbf{W}^{\sigma_{u_2}} \mathbf{c}_{2,t} + \mathbf{b}^{\sigma_{u_2}}) \tag{42}$$

### D.5    CALFADS LOSS FUNCTION

To construct the cost function of CaLFADS, we start from the evidence lower bound (ELBO)

$$
\begin{aligned}
P(\mathbf{x}) &\geq \mathbb{E}_{z,u_2,g_{2,0}\sim Q(z,u_2,g_{2,0}|x)}[\log P(x,u_1|\hat{x},\lambda)] \\
&\quad - D_{\mathrm{KL}}(Q(z,u_2,g_{2,0}|x)||P(z,u_2,g_{2,0})) \\
&= -\mathcal{L}_{ELBO}
\end{aligned}
\tag{43}
$$

where $\mathbf{x}$ denotes the observed data and $\hat{\mathbf{x}}$ denotes the reconstructed data.

With our model, this factorises as

$$
\begin{aligned}
-\mathcal{L}_{ELBO} &= \mathbb{E}_{\mathbf{z},\mathbf{u}_2,\mathbf{g}_{2,0}\sim Q(\mathbf{z},\mathbf{u}_2,\mathbf{g}_{2,0}|\mathbf{x})}[\log P(\mathbf{x}|\hat{\mathbf{x}})] \\
&\quad \mathbb{E}_{\mathbf{u}_2,\mathbf{g}_{2,0}\sim Q(\mathbf{z},\mathbf{u}_2,\mathbf{g}_{2,0}|\mathbf{x})}[\log P(\mathbf{u}_1|\lambda)] \\
&\quad - D_{\mathrm{KL}}(Q(\mathbf{u}_2,\mathbf{g}_{2,0}|\mathbf{x})||P(\mathbf{u}_2,\mathbf{g}_{2,0})) \\
&\quad - D_{\mathrm{KL}}(Q(\mathbf{z}|\mathbf{x})||P(\mathbf{z}))
\end{aligned}
\tag{44}
$$

Since $u_{1,t}$ is a continuous random variable that we are treating as if it came from an independent, homogeneous Poisson process with parameter $\lambda_t$, we approximate the log-likelihood with

$$
\log P(\mathbf{u}_1|\lambda) = \sum_t \log P(\mathbf{u}_{1,t}|\lambda_t)
\tag{45}
$$

$$
\approx \sum_t \mathbf{u}_{1,t}\log\lambda_t - \lambda - \mathbf{u}_{1,t} = -\mathcal{L}_{spike}
\tag{46}
$$

This approximation is very similar to the true log-likelihood of a Poisson distribution. The key difference is that the log-factorial term over the observations is replaced with the identity over the observations, i.e., $\log u_{1,t}! \approx u_{1,t}$. In deep learning applications, the log-factorial is often dropped entirely from the Poisson log-likelihood (see e.g., the Pytorch implementation and default setting[2]). It is sometimes replaced with a shifted log-gamma function $\log k! \approx \log\Gamma(k+1)$ or an approximation via Stirling's formula $\log k! \approx \frac{1}{2}\log 2\pi k$. Since Stirling's formula blows up near zero, and since log-gamma function has a local minimum between zero and one we decided against using these.

However, we observe that this term helps to enforce sparsity in the observations. Since this is desirable for inferred spike counts, we replace this term with what is essentially the $L_1$ norm of $u_{1,t}$, since by construction $u_{1,t}$ is forced to be positive.

The remaining terms of the ELBO were not approximated in any special way. As such we breakdown $L_{ELBO}$ as follows,

$$
\mathcal{L}_{recon} = -\mathbb{E}_{z,u_2,g_{2,0}\sim Q(z,u_2,g_{2,0}|x)}[\log P(x|\hat{x})]
\tag{47}
$$

$$
KL_1 = D_{\mathrm{KL}}(Q(z|x)||P(z))
\tag{48}
$$

$$
KL_2 = D_{\mathrm{KL}}(Q(u_2,g_{2,0}|x)||P(u_2,g_{2,0}))
\tag{49}
$$

The final cost function of CaLFADS becomes

$$
\mathcal{L}_{CaLFADS} = \mathcal{L}_{recon} + \mathcal{L}_{spikes} + w_{KL_1}\mathcal{L}_{KL_1} + w_{KL_2}\mathcal{L}_{KL_2}
\tag{50}
$$

where $w_{KL_1}$ and $w_{KL_2}$ are KL warmup terms that are typically used in training variational autoencoders. These terms slowly transition the model from a traditional autoencoder to a variational autoencoder, which helps to prevent pathological behaviour where the KL terms are minimised too early in training. This typically results in the model then generating trivial solutions. In our case, we use a staggered KL warm-up in which $\mathbf{w}_{KL_1}$ linearly increases from 0 to 1 over epochs 0-100, then $w_{KL_2}$ linearly increases from 0 to 1 over epochs 100-200. This allows the model to converge to a solution that overfits spikes to the fluorescence data first, then gradually regularises this proposed set of spikes by ensuring they are well explained by the low-dimensional latent factors.

---

[2]https://pytorch.org/docs/stable/generated/torch.nn.PoissonNLLLoss.html

# E  CALFADS DIRECTED ACYCLIC GRAPH

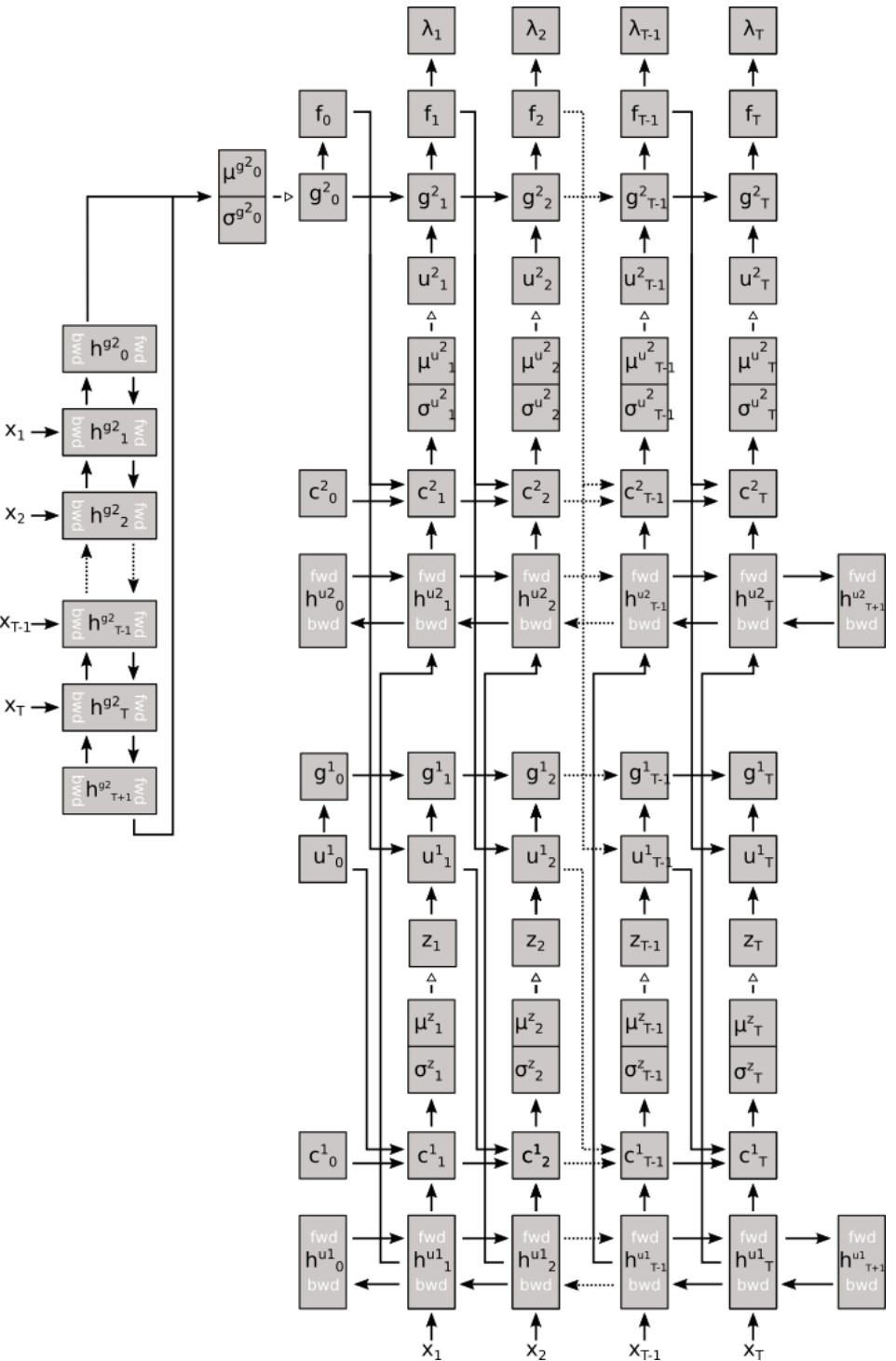

Figure S1: Directed acyclic graph for hierarchical model. Solid arrows denote deterministic mappings, open arrows denote sampling steps.

## F  CALFADS FORWARD METHOD PSEUDOCODE

---

**Algorithm 1:** Forward method for CaLFADS

---

1  Forward ;
    **Input**   : Fluorescence (dF/F) traces $x$ with dimensions $TxN$
    **Output:** Denoised dF/F $\hat{x}$, Inferred spike train $u_{1,t}$, Inferred dynamic factors $f_t$
2  $[h^{fwd}_{l=0,t=1:T}, h^{bwd}_{l=0,t=1:T}] = x_{t=1:T}$
3  **for** $t \leftarrow 1$ **to** $T$ **do**
4      **for** $l \leftarrow 1$ **to** $2$ **do**
5          $h^{fwd}_{l,t} = GRU(h^{fwd}_{2,t-1}, [h^{fwd}_{1,t}, E^{bwd}_{1,t}])$
6          $h^{bwd}_{l,T-t} = GRU(h^{bwd}_{2,T-t+1}, [h^{fwd}_{1,T-t}, h^{bwd}_{1,T-t}])$
7      **end**
8  **end**
9  $\mu_{g_{2},t=0} = Dense([h^{fwd}_{2,T+1}, h^{bwd}_{2,0}])$
10  $\sigma_{g_{2},t=0} = \exp(0.5 Dense([h^{fwd}_{2,T+1}, h^{bwd}_{2,0}]))$
11  $g_{2,t=0} \sim \mathcal{N}(\mu_{g_{2},t=0}, \sigma^2_{g_{2},t=0})$
12  $f_{t=0} = Dense(g_{2,t=0})$
13  **for** $t \leftarrow 1$ **to** $T$ **do**
14      $c_{1,t} = GRU(c_{1,t-1}, [h^{u,fwd}_{2,t}, h^{u,bwd}_{2,t}, u_{1,t-1}])$
15      $c_{2,t} = GRU(c_{2,t-1}, [h^{u,fwd}_{l,t}, h^{u,bwd}_{2,t}, f_{t-1}])$
16      $\mu_{u_{2},t} = Dense(c_{2,t})$
17      $\sigma^2_{u_{2},t} = \exp(Dense(c_{2,t}))$
18      $u_{2,t} \sim \mathcal{N}(\mu_{u_{2},t}, \sigma^2_{u_{2},t})$
19      $g_{2,t} = GRU(g_{2,t-1}, u_{2,t})$
20      $f_t = Dense(g_{2,t})$
21      $\lambda_t = \exp(0.5 Dense(f_t))$
22      $\mu_{z_{1},t} = Dense(c_{1,t})$
23      $\sigma^2_{z_{1},t} = \exp(0.5 Dense(c_{1,t}))$
24      $z_{1,t} \sim \mathcal{N}(\mu_{z_{1},t}, \sigma^2_{z_{1},t})$
25      $u_{1,t} = ReLU(\exp(Dense([z_{1,t}, f_t]) - 1))$
26      $g_{1,t} = AR1(g_{1,t-1}, u_{1,t})$
27      $\hat{x}_t = DyeKinetics(g_{1,t})$
28  **end**

---

## G CaLFADS LOSS FUNCTION PSEUDOCODE

---

**Algorithm 2:** Loss function for CaLFADS

---

1 CaLFADS Loss Function;

**Input** : Fluorescence (dF/F) traces $x$, reconstructed dF/F $\hat{x}$, Intensity function $\lambda$, Inferred spikes $u_1$, Posterior distribution $[\mu_{g_2,t=0}, \sigma_{g_2,t=0}, \mu_{u_{2,t}}, \sigma_{u_{2,t}}, \mu_{z_t}, \sigma_{z_t}]$

**Output:** Loss $\mathcal{L}$

2 $\mathcal{L}_{recon} = -\mathbb{E}_{z \sim Q(z|x)}[\log P(x|\hat{x})]$

3 $\mathcal{L}_{spike} = -\mathbb{E}_{z,g_2,u_2 \sim Q(z,g_2,u_2|x)}[\log(u_1|\lambda)]$

4 $\mathcal{L}_{KL_1} = w_1 D_{\mathrm{KL}}(Q(z|x)|P(z))$

5 $\mathcal{L}_{KL_2} = w_2 D_{\mathrm{KL}}(Q(g_2,u_2|x)|P(g_2,u_2))$

6 $\mathcal{L} = \mathcal{L}_{recon} + \mathcal{L}_{spike} + \mathcal{L}_{KL_1} + \mathcal{L}_{KL_2}$

7 return $\mathcal{L}$

---

## H    SYNTHETIC DATA GENERATION

As discussed in the main text, synthetic calcium fluorescence data was generated by embedding a Lorenz Attractor in the dynamics of a spiking neural network. Two models of calcium dynamics were then used to transform spikes to calcium transients, with additive emissions noise added to the resulting traces.

The Lorenz Attractor is a nonlinear dynamical system with 3 states $x_1, x_2, x_3$ commonly used to study chaotic dynamics. The dynamical system is defined by,

$$\frac{dx_1}{dt} = \sigma(x_2 - x_1) \tag{51}$$

$$\frac{dx_2}{dt} = x_1(\rho - x_3) - x_2 \tag{52}$$

$$\frac{dx_3}{dt} = x_1 x_2 - \beta x_3. \tag{53}$$

$$\tag{54}$$

This system was parameterised in its typical chaotic regime with $\sigma = 10, \beta = 8/3, \rho = 28$. These states were then normalised to have a mean of zero and a range of $[-1, 1]$.

The state of this system $\mathbf{x}$ was then randomly projected onto the firing rate of a population of $N = 30$ neurons,

$$\lambda_t = \exp(\mathbf{W}\mathbf{x}_t + \mathbf{b}) \tag{55}$$

with $\mathbf{W} \sim \mathcal{N}(0, \frac{1}{\sqrt{N}}\mathbf{I})$, and $\mathbf{b} = \mathbf{1}$. Spike counts in time-bin $t$ for neuron i $s_i$ were then sampled using a Poisson distribution

$$s_{i,t} \sim Poisson(\lambda_{i,t}\Delta t) \tag{56}$$

where $\Delta t$ is the width of a time-bin. For our simulations, we used $\Delta t = 0.1s$.

Calcium fluorescence traces were then modelled in one of two ways. The first method modelled calcium concentration in neuron i $c_i$ as an exponentially decaying variable with time constant $\tau = 0.3$ and perturbations by spikes

$$\frac{dc_i}{dt} = \frac{-c_i}{\tau} + s_i. \tag{57}$$

Fluorescence was then modelled by adding white emissions noise with standard deviation $\sigma_F = 0.2$.

$$F_t = c_t + \epsilon_t \qquad \epsilon \sim \mathcal{N}(0, \sigma_F) \tag{58}$$

The second method for generating synthetic calcium traces added an intermediary step between calcium influx and observed fluorescence. As previously, spikes were integrated with a slow-varying exponentially decaying variable $c$ as described in Equation 57. A hill function was used to capture the nonlinear binding kinetics of calcium to the indicator dye,

$$d_t = \frac{c_t^n}{1 + \gamma c_t^n} \tag{59}$$

with hill coefficient $n = 2$, and $\gamma = 0.0001$. These parameters were chosen based on parameter fits in Deneux et al. (2016). Finally white noise was added to these resulting traces to provide synthetic fluorescence traces,

$$F_t = d_t + \epsilon_t \qquad \epsilon \sim \mathcal{N}(0, \sigma_F) \tag{60}$$

# I  CaLFADS EXAMPLE OUTPUTS AND SCATTER PLOTS ON SYNTHETIC BENCHMARK DATASETS

## I.1  LORENZ ATTRACTOR: LINEAR CALCIUM MODEL

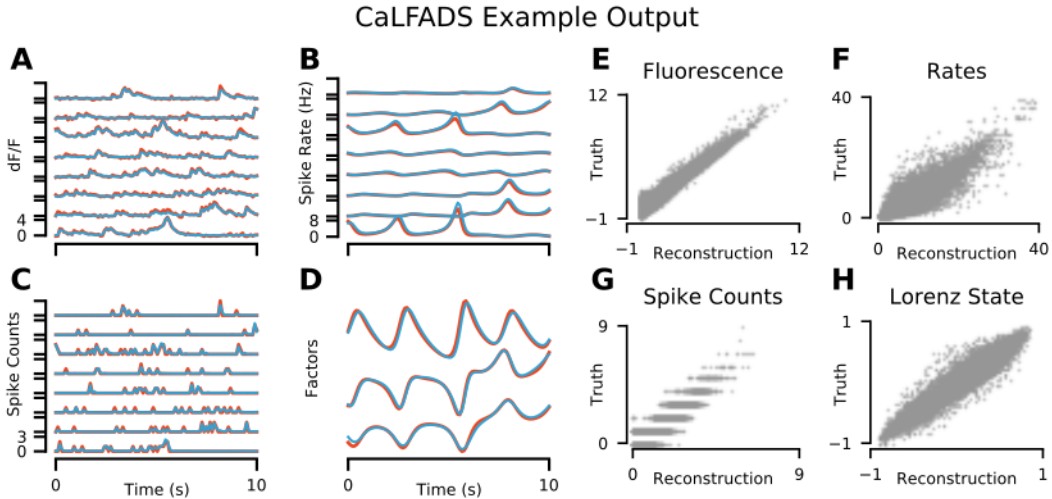

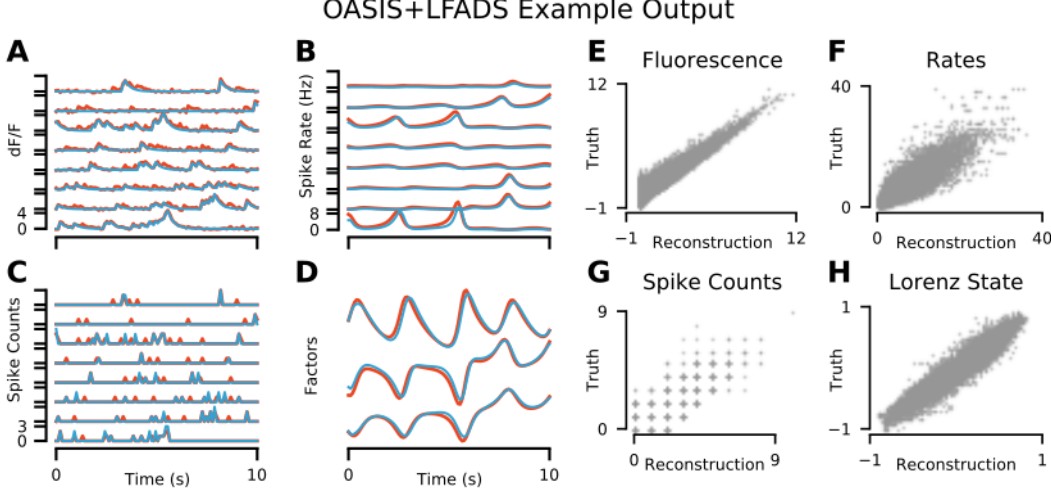

Figure S2(Top), S3(Bottom): Example traces of A) Observed Fluorescence, B) Spike Rates, C) Spike Counts, D) Dynamic Factors. Red: Ground-truth, Blue: Reconstructed. Scatter plots comparing reconstructed and ground-truth of E) Observed Fluorescence, F), Spike Rates, G) Spike Counts, H) Lorenz State

Figure S2 shows examples of the performance of CaLFADS in reconstructing the fluorescence traces (Fig S2A), spike rates (Fig S2B), spike counts (Fig S2C) and Lorenz attractor states (Fig S2D) when using a linear model of synthetic calcium dynamics. Visually, it can be seen that the model achieves a very close fit to the fluorescence traces, spike rates, and Lorenz dynamics. The model also captures spike-timing, with these spike trains appearing smoothed.

For comparison, Figure S3 show examples of OASIS+LFADS reconstructions. A key difference is that the spike counts are not smooth as spike counts are discretised from the deconvolved transient amplitudes, and there is greater variance in reconstruction quality at high spike rates as shown by

the scatter plot in Figure S3F. Nevertheless, OASIS+LFADS also provides a good fit to fluorescence traces, spike rates, spike counts and lorenz dynamics.

## I.2 Lorenz Attractor: Nonlinear Calcium Model

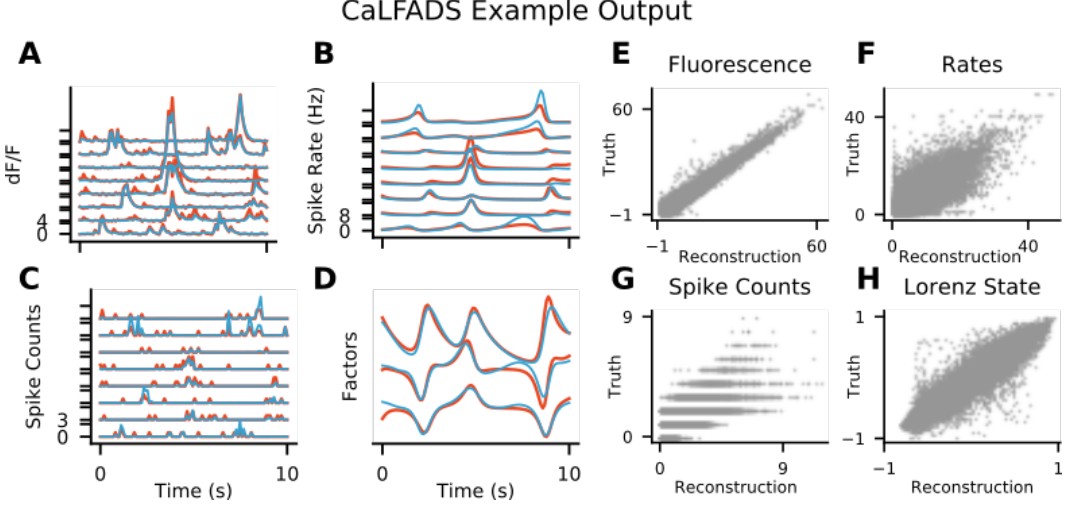

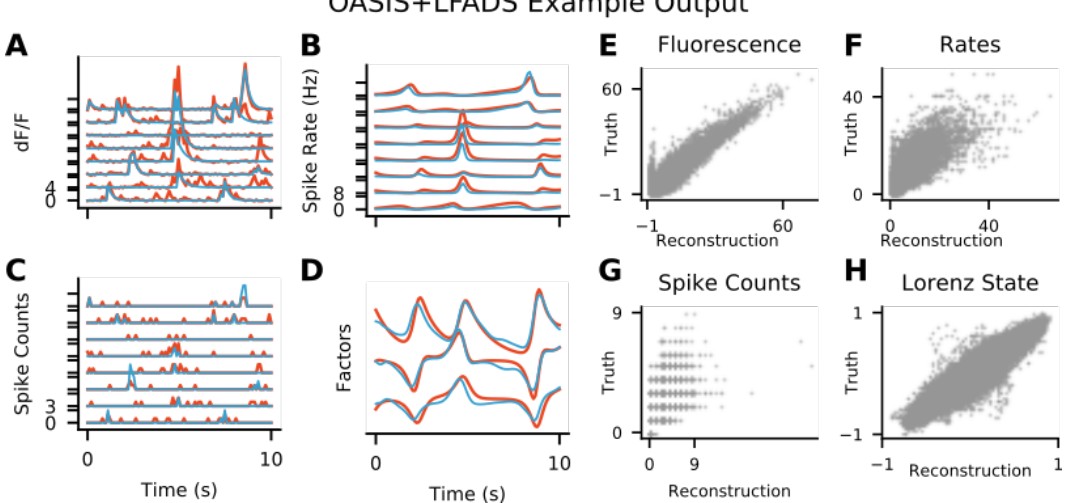

Figure S4(Top), S5(Bottom): Example traces of A) Observed Fluorescence, B) Spike Rates, C) Spike Counts, D) Dynamic Factors. Red: Ground-truth, Blue: Reconstructed. Scatter plots comparing reconstructed and ground-truth of E) Observed Fluorescence, F), Spike Rates, G) Spike Counts, H) Lorenz State

Figure S4 shows examples of the performance of CaLFADS in reconstructing the fluorescence traces (Fig S4A), spike rates (Fig S4B), spike counts (Fig S4C) and Lorenz attractor states (Fig S4D) when using a nonlinear model to generated synthetic calcium traces. As with the linear calcium model, it can be seen that the model achieves a close reconstruction of to the Lorenz dynamics. However, the reconstructions of other network variables have deteriorated to a much greater extent. A similar trend can be seen in Figure S5 for OASIS+LFADS.

## J    MOUSE VISP DATASET, DECODING NETWORK AND EXAMPLE FACTORS

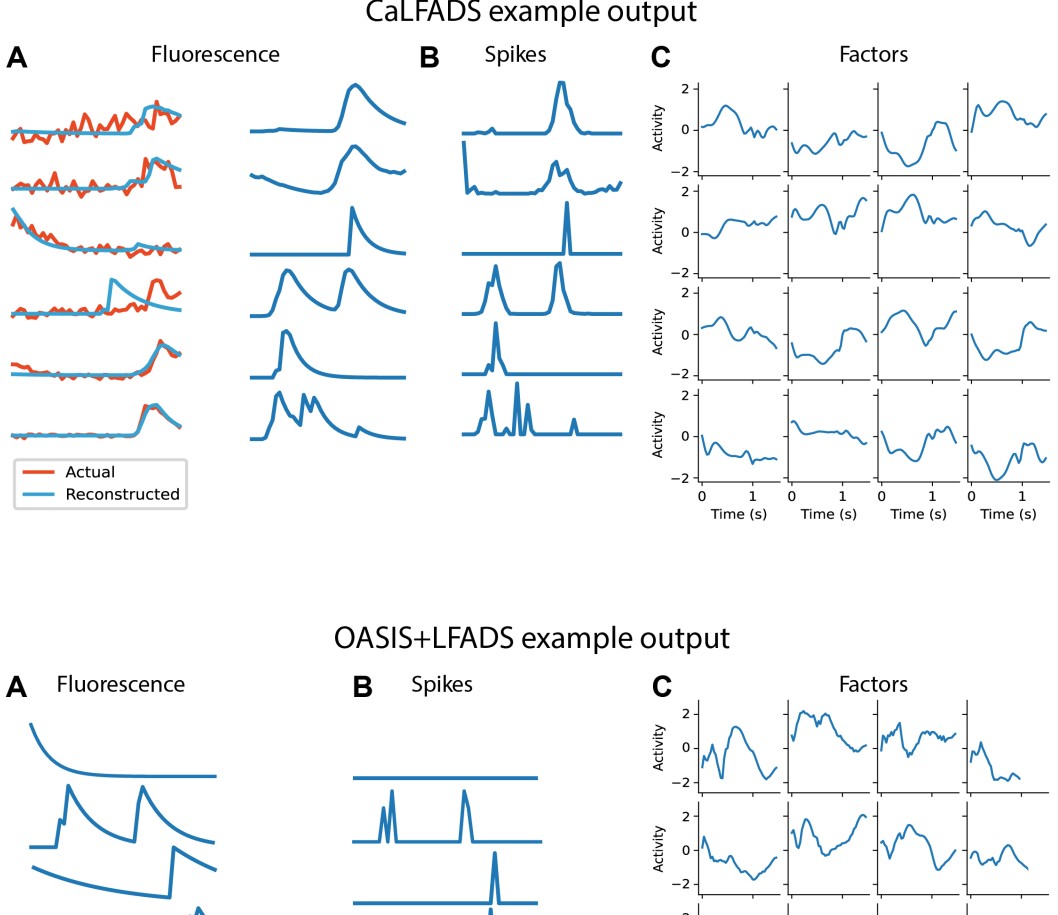

Figure S6(Top), S7(Bottom): A) 6 samples of actual and reconstructed fluorescence traces (S6A, left column) and 6 samples of fluorescence traces (S6A right column and S7A), B) 6 samples of inferred spike count traces corresponding to the fluorescence traces in A (right column, for S6A), C) 12 samples of inferred latent factor traces.

### J.1    MOUSE VISP DATASET

The 2-photon calcium imaging data used here is part of the OpenScope project dataset (Anonymous, 2018), collected by the Allen Institute for Brain Science (AIBS) in Seattle, WA. All animal procedures were approved by the Institutional Animal Care and Use Committee at the AIBS. Neuronal activity was recorded in head-fixed, awake Cux2-Cre mice on a running wheel (de Vries et al., 2020) (Fig 3A) expressing the calcium indicator GCaMP6f (Chen et al., 2013) in layer 2/3 pyramidal cells of VisP (Fig 3B). The mice were first habituated to a repeating, expected stimulus over 6 days, after which unexpected trials were introduced. The stimuli were adapted from (Homann et al., 2017). In each expected trial, 4 consecutive sets of 30 Gabor patches appeared in sequence for 300 ms each, followed by 300 ms of grey screen (A, B, C, D, grey). For each set, the locations and sizes of the

Gabor patches were held constant within a session. However, within each trial, each Gabor patch's orientation was sampled from a von Mises distribution, with trial mean sampled from {0, 45, 90, 135}° and standard deviation 0.25°. In expected trials, occurring 10% of the time, the D set was replaced with a distinct set E, with its own locations and sizes. In addition, the Gabor patch orientations for E sets were sampled from a von Mises distribution with mean shifted positively by 90° (Fig 3C). Processed fluorescence (dF/F) traces were extracted from the calcium imaging recordings for the putative neurons identified, as described in (de Vries et al., 2020).

## J.2 MOUSE VISP DATA: EXAMPLE OUTPUTS

Figures S6 and S7 show samples of actual fluorescence traces (A), the corresponding inferred spike count traces (B), and inferred latent factors (C) generated as the output of the CaLFADS and OASIS+LFADS models, respectively. Figure S6A additionally shows examples of reconstructed and actual fluorescence traces (A, left column). As can be seen in the figures, the inferred latent factors of CaLFADS are smoother and less noisy compared to those of the OASIS+LFADS. This is important as the quality of the inferred factors is crucial for decoding purposes, such as the classification task described in Appendix J.3.

## J.3 MOUSE VISP DECODING NETWORK AND EXAMPLE FACTORS

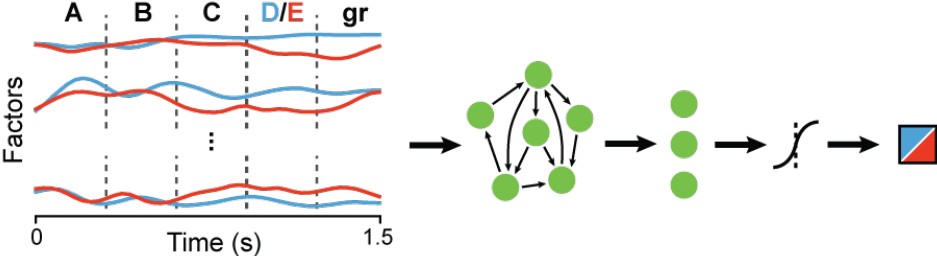

Figure S7: Schema of the non-linear model trained to decode expected vs. unexpected stimulus trials. Example latent factors inferred by CaLFADS are plotted for an expected (blue) and an unexpected (red) trial. Each stimulus frame (A, B, C, D/E, grey) is labelled, and its onset and offset in the trial are marked with dotted lines. Latent factors are passed through GRU and linear modules (green), followed by a sigmoid decision function.

Figure S7 shows a schematic of the non-linear model used to decode from the latent factors whether a trial contained an expected or unexpected frame (decoding performances reported in Fig 3D). In short, the non-linear decoder comprised of a single-layer GRU, followed by a linear layer and a sigmoid decision function. We show an example set of latent factors (Figure S7 left) inferred in expected (blue) and unexpected (red) trials. From these, one can see that CaLFADS was able to infer smooth latent factors from the calcium imaging data that did not show signs of over-fitting to noise.

