# OpenReview forum: "CaLFADS: latent factor analysis of dynamical systems in calcium imaging data"
_ICLR.cc/2021/Conference — Reject_

### Official Review · AnonReviewer2 · 2020-10-25
**Application of sequential hierarchical autoencoders to calcium imaging in neuroscience**

**Rating:** 4
**Confidence:** 4

**Review:**

------------- UPDATE FOLLOWING DISCUSSION -------------

The authors were responsive and suggested a series of updates to address the concerns raised here. I find the inclusion of a theory section valuable. Overall, though, my stance did not change: I’d like to encourage the authors to pursue their idea, but I don’t think the paper is ready for acceptance yet. I’d recommend focusing on two aspects:
(i) Simulations and numerical results are especially confusing. Please consider a major revision in light of the comments raised by different reviewers.
(ii) I find the application to fMRI interesting and potentially impactful. If the authors can include such analysis, I think it’d improve the paper significantly. Being a completely different experiment, it would also alleviate the concerns on numerics. (significance, effect size, stability)

------------- END OF UPDATE -------------

Summarizing the dynamics observed via high-dimensional recordings of neuronal activity is an important problem in neuroscience. Imaging the calcium dynamics of neurons via calcium binding fluorescent molecules, rather than recording the spiking electrophysiological activities, enables higher throughputs and has become popular. This, however, creates the additional step of inferring the underlying spikes from the smooth calcium activity recordings. This paper addresses the problem of inferring the latent dynamics directly from calcium activity recordings by jointly optimizing for both steps with a sequential ladder variational autoencoding architecture.

The paper demonstrates (somewhat small) improvements over existing methods, and it may be useful for neuroscientists. However, I do not support the acceptance of this submission due to the following major concerns:

--- Limited methodological novelty: The approach seems to be a direct application of a particular sequential hierarchical autoencoder to this particular calcium-imaging problem. There is limited methodological discussion. In this sense, the manuscript may be a better fit for a neuroscience journal.
--- Focusing on a single domain: The experiments cover a narrow domain for an application-focused paper. The authors do not provide example use cases beyond calcium imaging in neuroscience.
--- Comparison with the baseline method (OASIS+LFADS): OASIS was designed for fast, online spike inference from calcium imaging data.
     a) How much time is needed for OASIS+LFADS vs CaLFADS? If OASIS+LFADS is faster, could more complicated settings/architectures be used for OASIS/LFADS to improve performance? (e.g, lag window size in OASIS, recurrent network size in LFADS)
     b) The OASIS paper mentions different working modes, where only some of the possible parameters are optimized. How was OASIS run in this manuscript?
--- Exposition: The paper focuses on neuroscience-oriented readers only, and may not appeal much to the more general ML crowd. For instance, I thought it would help a lot to display the formal probabilistic model using equations.

Other concerns:
--- The paper refers to the different dynamics within the hierarchy as higher-/lower-order dynamical systems. The order of a dynamical system has an established definition in the literature already. Unless the authors cite previous such usage, I suggest changing this potentially confusing terminology.
--- C. Pseudocode: (i) T\times N, (ii) lines 5-6, right hand side doesn’t depend on l. (iii) Variables aren’t properly defined. e.g., What is E_{l,t}^{bwd}?
--- What does 14 represent in t_{14} in Table 1?

---

> ### Author Response · Authors · 2020-11-21
> **Response to reviewer 2**
>
> Thank you for you review. It highlighted to us that we needed to make it clearer what CaLFADS offers to machine learning practitioners and neuroscientists.
>
> **1. The paper demonstrates (somewhat small) improvements over existing methods**: We respectfully disagree with one of Reviewer 2’s key justifications for our low score, namely that the performance gain offered by CaLFADS is insufficient. We would like to stress the point that the improvement in performance by CaLFADS over other methods is statistically significant. Furthermore, to reinforce our claim that CaLFADS learns more meaningful latent factors, we retrained all models on the mouse V1 data with multiple seeds. To our surprise and fortune, the performance gain only increased (CaLFADS: 0.871, OASIS+LFADS: .810, Gaussian-LFADS: 0.500, PCA: .815). We believe this is a valuable improvement over previous methods, and would like further details on what level of improvement Reviewer 2 would consider sufficient for acceptance.
>
> **2. Limited methodological novelty:** Can reviewer 2 elaborate on why they feel the development of this sequential hierarchical autoencoder is of limited novelty? To our knowledge, and as Reviewer 3 pointed out, this is the first time a hierarchical VAE has been applied to sequential data. If Reviewer 2 can share examples of related work that we have missed in this area we would be grateful. Moreover, as Reviewer 3 also recognised, it is non-trivial to train hierarchical autoencoders, since it is common for representations to collapse into the shallowest layer of the hierarchical latent space. CaLFADS does not suffer from this problem due to the appropriate factorization and splitting of the latent variable representation in the network, as suggested by the VLAE architecture (which had not been extended to sequential data before), and a staggered KL warmup that allowed shallower latent variables to be learned ahead of deeper latent variables. For these reasons we believe we demonstrate sufficient methodological novelty. We recognize that we are building directly on LFADS, but we would argue that building on the progress of others is precisely how machine learning research typically operates.
>
> **3. Focusing on a single domain**: We disagree with Reviewer 2’s assessment that the focus on calcium imaging is a weakness of this paper. The vast majority of machine learning papers with empirical results apply methods to a single domain (images, videos, speech, single language text). ICLR expressly calls for papers studying “applications in audio, speech, robotics, neuroscience, computational biology, or any other field”. ICLR has accepted a number of papers in neural data analysis in previous years, see e.g., https://openreview.net/forum?id=rklr9kHFDB, https://openreview.net/forum?id=r1VVsebAZ and https://openreview.net/forum?id=SkloDjAqYm for a calcium imaging paper. We argue in particular that two-photon calcium imaging presents a problem of interest to the broader machine learning community since it is a situation where there is clearly a need to separate hierarchies of dynamics.
>
> **4. How much time is needed for OASIS+LFADS vs CaLFADS? If OASIS+LFADS is faster, could more complicated settings/architectures be used for OASIS/LFADS to improve performance? (e.g, lag window size in OASIS, recurrent network size in LFADS):** It takes 3x longer to train CaLFADS. Increasing the layer widths of LFADS in OASIS+LFADS does not seem to have an effect on its performance in reconstructing the Lorenz attractor or on decoding performance on V1 dataset.
>
> **5. The OASIS paper mentions different working modes, where only some of the possible parameters are optimized. How was OASIS run in this manuscript?** OASIS was run without knowledge of the ground-truth AR(1) parameters. As such it was used to infer the amplitude of fluorescence change caused by single spikes and the fluorescence decay time constant.
>
> **6. No equations for probabilistic model:** We apologise for this omission as we felt the pseudocode was sufficient. Clearly it has not been as reviewers 1 and 3 have also remarked on this. We have added the mathematical form of the generative model in appendix D.  Please see  https://pdfhost.io/v/gdy6BjPtq_CaLFADS_new_pagespdf.pdf for a draft.
>
> **7. Confusing terminology**: Reviewer 1 also commented on this, and we recognize now that it is a problem. We acknowledge that the terminology could be a source of confusion, especially since the order of ODE systems is already defined. Here, we were looking to describe the relative ‘depth’ of the latent dynamics in the hierarchy, to refer to how close these dynamics were to the observed data. We are happy to change our terminology to prevent further confusion, for example we could change ‘higher-order’ to ‘deeper-level’ and lower-order’ to ‘shallower-level’
>
> **8. What does 14 represent in t_{14} in Table 1? 14 is the degrees of freedom in the t-distribution for a paired t-test.

---

> > ### Comment · AnonReviewer2 · 2020-11-24
> > **Thanks for the response. Adding equations was useful.**
> >
> > I agree with AnonReviewer1’s response. The uncertainty (sem) estimates make one of the two results (original, updated) highly unlikely. Without explaining the underlying reason (e.g., a bug in original calculations, systematically too small uncertainty estimates in both versions), it’s hard to be convinced either way. Also, please note that statistical significance and effect size are not the same thing.
> >
> > Application: I think broadness of the application domain should be judged based on multiple factors, (e.g., effect size, methodological novelty, extensiveness of benchmarking experiments, popularity of the domain), and not in isolation. I do not necessarily expect to see experiments/results in other domains. But, for instance, it would improve the paper if the authors identified 1-2 example problems in other domains, and drew analogies between their model parameters and the variables in the calcium imaging case. The single sentence on other applications (“…our approach may be applicable to other life sciences domains where the observed variables have relatively slow dynamics, and thus, could serve as a basis for the construction of future analysis tools”) is more mystifying than motivating.
> >
> > Methodological novelty: There is no question that the work is not repetitive in its application domain. However, in my opinion, the original submission did not make the case for methodological novelty at all. Adding equations helped a lot in this sense - thanks. I still do not think that the methodological difference goes significantly beyond relatively straightforward adaptation. (To be clear, there is methodological novelty, but I find it limited.) On the other hand, methodological novelty is certainly not the only aspect that makes a paper appealing.
> >
> > Terminology: Thanks - yes, ’deeper-/shallower-level’ makes it much clearer for me.

---

> > > ### Author Response · Authors · 2020-11-25
> > > **Second response to Reviewer 2**
> > >
> > > Thank you for taking the time to provide additional feedback on our updated submission.  We provide clarifications below, in addition to information about a second set of modifications made to the manuscript.
> > >
> > > **Need to further explain the change in mouse VisP results:** We understand yours and Reviewer 1’s concern about the difference between our original and updated results for Fig. 3. We had originally performed the analysis using only one seed for training the models and training recurrent neural networks as the classifiers on 50 folds of the VisP data. At that point we obtained a recall of 0.805 ± 0.008 with OASIS+LFADS and 0.826 ± 0.005 with CaLFADS.
> > >
> > > Post-submission, we re-ran our experiments on a GPU cluster in order to collect more results faster. Indeed, based on the feedback, we decided it was important to further verify that our results held over different sets of latent factors. Therefore, we trained the models with 4 different random seeds instead of 1 (then evaluated recall for each model on 200 folds of the VisP data for each set of inferred latent factors). The GPUs on the cluster happened to have different specifications than the GPU on the first author’s laptop, and thus ran different versions of Pytorch with different CUDA drivers. Unfortunately, seeded processes are not guaranteed to be reproducible across pytorch and CUDA versions (examples can be found on Pytorch forums). As a result, we did not end up reproducing our original result on the GPU cluster. Furthermore, we were surprised to find that all four seeds led to higher performances across the board for CaLFADS:
> > >
> > > Our new results on the 4 seeds were (mean ± SEM):
> > >
> > > **Seed 1:** 0.880 ± 0.006 (CaLFADS)
> > > **Seed 2:** 0.865 ± 0.005 (CaLFADS)
> > > **Seed 3:** 0.859 ± 0.009 (CaLFADS)
> > > **Seed 4:** 0.881 ± 0.010 (CaLFADS)
> > > **Pooled statistics reported:** 0.871 ± 0.004
> > >
> > > We also made sure to rerun all OASIS+LFADS models on the new hardware. Here, our results were similar to the original results.
> > >
> > > **Seed 1:** 0.815 ± 0.011 (OASIS+LFADS)
> > > **Seed 2:** 0.813 ± 0.011 (OASIS+LFADS)
> > > **Seed 3:** 0.829 ± 0.009 (OASIS+LFADS)
> > > **Seed 4:** 0.784 ± 0.009 (OASIS+LFADS)
> > > **Pooled statistics reported:** 0.810 ± 0.005
> > >
> > > We believe that the boost in our CaLFADS results are likely due to the inferred latent factors based on the original random seed being slight outliers in terms of being directly suited for our downstream classification task. The updated results however do appear robust, and show high consistency across seeds. The original results showed that even “less suited” factors still provide improved performance over OASIS+LFADS.
> > >
> > > Of course, it is indeed important to consider both significance and effect size. We believe that the breakdown of results here supports the conclusion that CaLFADS robustly provides an approximate improvement of about 6% as compared to OASIS+LFADS (or also PCA on the dF/F) on our task.
> > >
> > > **Application:** We understand that our statement that the CaLFADS approach could be applicable to other life sciences domains was not specific enough. We have expanded on it to be clearer about what we have in mind, bringing up two examples (fMRI for neuroscience and insurance claim modelling for domains beyond) while indicating how these datatypes are analogous to the calcium data and thus amenable to a CaLFADS-like approach.

---

### Official Review · AnonReviewer1 · 2020-10-28
**Potentially very interesting, but I lack clarity on certain key issues**

**Rating:** 5
**Confidence:** 4

**Review:**

This paper, as I understand it, extends a very nice method for Latent Factor Analysis of Dynamical Systems (LFADS) from spike train data to calcium imaging data.  I am familiar with the LFADS paper, as well as much of the literature on calcium imaging for spike inference, and latent dynamical systems more generally.  I am in no way an expert on anything deep learning, nor nonlinear dynamics such as Lorenz attractors.  I had trouble building an intuition about the model of neural dynamics, the approach, and the results, possibly due to the presentation style, which I was not used to, and I explain below.

1. I am very comfortable with equations, including explicit generative models of the calcium, and inference/encoder models as well.  The LFADS paper, for example, explained both quite clearly.  In this paper, however, I could not find the relevant equations, other than those for the synthetic data generator. I struggled to understand therefore, precisely what was the implicit model, which was informally described in the 2nd paragraph of the paper.  Similarly, the encoding model was described informally in section 2, and with Figure 1, and Table S1 and Figure S1 helped, but I really wished there were equations.   The LFADS paper has equations that I understand, so I guessed one possibility was that this generative model generalized that by adding another random variable "on top" of the spikes to characterize the calcium activity. Then I would have guessed that the encoder/inference models would be modified similarly, but I could not tell.  Algorithm 1 helped for the forward method, but even there some of the notation was not clear (e.g., what precisely does 'Dense' mean?).

2. The discussion around lower-order and higher-order dynamics confused me a bit as well, as I could not find formal definitions of those terms.  I guessed that lower-order dynamics referred essentially to first-order markov chains, and second-order dependencies counted as higher-order, but that logic didn't quite work throughout.  If parsing the lower vs higher order dependencies is a key desiderata of this method, it would have been nice to explicitly see the approach doing that in the results.

3. The results in Table 1 confused me a bit.  Why are three significant digits shown for R^2, there were 20 different seeds, so precision down to 1/1000 does not seem possible.  Also, I lack intuition as to what the different true estimated latent states look like.  Do I care about 0.005 better R^2? I would think not.  t-tests are not obviously valid tests here either.

4. Figure 2 was very nice and clear and helpful.  It clearly shows that on the synthetic data, CaLFADS does something closer (in some qualitative, though not quantiative) sense, than OASIS+LFADS.  I did wonder, however, OASIS is a fully online algorithm, whereas LFADS and CaLFADS are batch algorithms.  So, if I understand correctly, OASIS is it a large disadvantage because it cannot use data after each spike to help with the inference.  This is particularly important for fast dynamics, because the post-spike effect can be much more informative than the pre-spike information.  So, I would think that a more fair approach would be to use a batch, rather than online, comparison.

5. Figure 3 was clear enough about recall.  I wonder, from the experimentalist perspective, if they can get 0.8 simply by operating on the raw data, and can get 0.82 by running this complicated model, would they?  I guess the answer depends on how difficult and time-consuming it is for them and their computer to run this, but no information is provided about that which I can find.  I looked through the code briefly, and it seems maybe easy enough, but I didn't see a tutorial for using it on my data.  Also, one of the notebooks somewhat un-anonymizes you because it includes direct paths on your computer, rather than relative paths :)

To provide some concrete recommendations:

1. I'd like more equations, the more the merrier for me, could be in the appendix.
2. I'd like figures showing the data, both real and synthetic, as well as the estimated latent states, so we can understand what the summary of the errors means.
3. I'd like more explicit clarity on any changes you made to LFADS other than adding a layer on top, what precisely those changes were, what precisely motivated those changes, and whether they were worth it.

In conclusion, I think this paper adds a relatively incremental addition to LFADS, which could potentially be interesting to neuroscientists, but I failed to clearly understand the ingenuity of the approach, or the impact of the empirical results.

===============================================================
==== updates after rebuttal period =================================

1. The authors added more equations, I now understand the methodological contributions required substantial effort.  The VAE/etc. work is outside my area of expertise.
2. The authors added some figures showing both real and synthetic data. I've looked through them, and the statistics carefully.  Personally, reading through all of this, I would use PCA+OASIS rather than this new technique.  I understand R^2 is higher for this technique, but looking at the data, without actual confirmation of any of the inferences. There exists data with joint calcium imaging and ephys one can use to calibrate and evaluate methods, those are the best data to use for such purposes.  In the absence of those data, we are guessing.  My guess is that the OASIS+LFADS output looked better than the CaLFADS output. R^2 is a funny metric, especially when we are trying to compare spike trains, because translations utterly break R^2, unless the output is smooth, in which case, it does not.  It looked to me that CaLFADS had a smoother output than OASIS+LFADS, which could make R^2 higher, but is not actually what I typically want when analyzing calcium imaging data. Indeed, if we were ok with smoothing, we could simply operate on the calcium imaging data itself, without worrying about spike train inference.
3. I got more clarity on the differences between LFADS and CaLFADS, though again, this is outside my area of expertise.

---

> ### Author Response · Authors · 2020-11-21
> **Response to reviewer 1**
>
> Thank you for your review. We appreciate your interest in our paper, and the directness of your criticisms. Based on your criticisms, we are keen to express our confidence that CaLFADS is an exciting and distinct contribution to both machine learning and neuroscience. Here we address your concerns in more detail.
>
> **1. No equations for the model**: We apologise for this omission as we felt the pseudocode was sufficient. Clearly it has not been as reviewers 2 and 3 have also remarked on this. We have added the mathematical form of the generative model in appendix D.  Please see https://pdfhost.io/v/gdy6BjPtq_CaLFADS_new_pagespdf.pdf for a draft.
>
> **2. Lower-order and higher-order dynamics terminology was confusing.** Reviewer 2 also commented on this. We acknowledge that the terminology could be a source of confusion, especially since the order of ODE systems is already defined. Here, we were looking to describe the relative ‘depth’ of the latent dynamics in the hierarchy, to refer to how close these dynamics were to the observed data. We are happy to change our terminology to prevent further confusion, for example we could change ‘higher-order’ to ‘deeper-level’ and lower-order’ to ‘shallower-level’
>
> **3. Precision of R^2.** R^2 is computed as a goodness-of-fit for the VAE reconstruction to the validation dataset. This is a comparison of  # seeds x # trials x # time-steps x # cells = 1.17 x 10^7 fluorescence values. This is more than enough to report 3 significant figures.
>
> **4. OASIS is a fully-online algorithm, whereas CaLFADS is a batch algorithm. It would be a fairer approach to use a batch rather than online comparison.** To clarify, we actually used OASIS offline in a batch setting. Furthermore, OASIS has been demonstrated to be a very robust method for inferring spikes for single cells (Pachitariu et al., 2018), even compared with batch-only algorithms. We don’t agree that there is much value in comparing to other spike inference algorithms given their relatively similar levels of performance. Can Reviewer 1 elaborate on why they think it is justified?
>
> **5. I wonder, from the experimentalist perspective, if they can get 0.8 simply by operating on the raw data, and can get 0.82 by running this complicated model, would they?** While we feel that this is somewhat of a leading question, we will attempt to respond to it head-on. Yes, since we think CaLFADS is better at conserving complex information in the latent factors that are easier to decode. This makes it worthwhile for experimentalists to use to understand the neural mechanisms underlying the behavioural responses they observe. Furthermore, to reinforce our claim that CaLFADS learns more meaningful latent factors, we retrained all models with multiple seeds. To our surprise and fortune, the performance gain only increased (CaLFADS: 0.871, OASIS+LFADS: .810, Gaussian-LFADS: 0.500, PCA: .815).
>
> **6. I'd like more explicit clarity on any changes you made to LFADS other than adding a layer on top, what precisely those changes were, what precisely motivated those changes, and whether they were worth it… I think this paper adds a relatively incremental addition to LFADS.** Respectfully, we would like to propose to the reviewer that this is an unfair critique of CaLFADS. First, we note that CaLFADS is at least as different from LFADS as LFADS is to DRAW (https://arxiv.org/abs/1502.04623). The difference between LFADS and DRAW is an additional layer ‘on-top’ to infer the initial conditions of the generator, and a Poisson likelihood term for the observed spikes. The difference between CaLFADs and LFADS are additional layers to infer the timing of spikes for reconstructing calcium fluorescence traces and a hierarchical likelihood term. As it is common for representations to collapse into a single layer of the hierarchy, these layers had to be structured in such a way as to ensure that calcium dynamics are separated from computational dynamics. As Reviewer 3 acknowledged, building hierarchies of latent variables in neural networks is a non-trivial task. Furthermore, as Reviewer 3 also mentioned, this is the first time that a hierarchical VAE has been successfully applied to sequential data at all. Additionally, we found that it was necessary to use a staggered KL warmup to appropriately separate hierarchies of latent variables. To our knowledge, this is also the first time a staggered KL warmup has been used in hierarchical inference. Since deviations from the ELBO in training VAEs (including KL weights, KL annealing strategies) are an active research topic, we think this is also an interesting and novel contribution. Therefore, altogether, though we agree with the reviewer that we are building very directly on LFADS, we feel that our paper makes a number of non-trivial extensions that are novel to at least the same degree as in other similar papers, and importantly, unique within the field more broadly.

---

> > ### Comment · AnonReviewer1 · 2020-11-24
> > **Thanks, though still confused about errorbars**
> >
> > Thanks for adding the equations and explaining the innovations, which were not clear to me in the previous version of the manuscript.
> >
> > I remain concerned, however, about the real data example.  Specifically, previously the recall you got was 0.82, now it is 0.87.  In the revised paper, the errorbars (which are the sem, although I'm not sure precisely what is being averaged over, trials? cells? how many?) are 0.004.  Something seems wrong, because two runs differ by 0.05, which is over an order of magnitude larger than the reported errorbars.  This is also related to my previous claim about the precision of the reported numbers.  I now understand the sample size considerations, but this is a randomized algorithm. Note that if the recall shifted 0.05 in one direction one time, there is relatively little evidence that it would not shift 0.05 in the other direction the next time, which would make it worse than simply running PCA.
> >
> > I appreciated the figures showing the simulated data, perhaps if you include figures showing the real data, it would be more compelling.  I guess I am asking you to show me something that as an experimentalist, I'd think: yes, I definitely want that.  Or, show me that this method does not yet get that, and that's ok too.
> >
> > In my view, the ML field in general has a highly problematic view on empirical results, nearly all of which are cherry-picked.  Without pre-specifying the details of the experiments, there are enough degrees of freedom to essentially always find a dataset/trial/etc. in which a new algorithm outperforms previous algorithms.  Fundamentally, this whole approach is unimpressive, so this is a criticism of the field more than the paper.  All that said, given the essentially unlimited flexibility in finding a compelling numerical experiment, I'd expect it to be very convincing.  Maybe it is already and I fail to see it.

---

> > > ### Author Response · Authors · 2020-11-25
> > > **Second response to reviewer 1**
> > >
> > > Thank you for taking the time to provide additional feedback on our updated submission.  We provide clarifications below, in addition to information about a second set of modifications made to the manuscript.
> > >
> > > **Need to further explain the change in mouse VisP results:** We understand yours and Reviewer 2’s concern about the difference between our original and updated results for Fig. 3. We had originally performed the analysis using only one seed for training the models and training recurrent neural networks as the classifiers on 50 folds of the VisP data. At that point we obtained a recall of 0.805 ± 0.008 with OASIS+LFADS and 0.826 ± 0.005 with CaLFADS.
> > >
> > > Post-submission, we re-ran our experiments on a GPU cluster in order to collect more results faster. Indeed, based on the feedback, we decided it was important to further verify that our results held over different sets of latent factors. Therefore, we trained the models with 4 different random seeds instead of 1 (then evaluated recall for each model on 200 folds of the VisP data for each set of inferred latent factors). The GPUs on the cluster happened to have different specifications than the GPU on the first author’s laptop, and thus ran different versions of Pytorch with different CUDA drivers. Unfortunately, seeded processes are not guaranteed to be reproducible across pytorch and CUDA versions (examples can be found on Pytorch forums). As a result, we did not end up reproducing our original result on the GPU cluster. Furthermore, we were surprised to find that all four seeds led to higher performances across the board for CaLFADS:
> > >
> > > Our new results on the 4 seeds were (mean ± SEM):
> > >
> > > **Seed 1:** 0.880 ± 0.006 (CaLFADS)
> > > **Seed 2:** 0.865 ± 0.005 (CaLFADS)
> > > **Seed 3:** 0.859 ± 0.009 (CaLFADS)
> > > **Seed 4:** 0.881 ± 0.010 (CaLFADS)
> > > **Pooled statistics reported:** 0.871 ± 0.004
> > >
> > > We also made sure to rerun all OASIS+LFADS models on the new hardware. Here, our results were similar to the original results.
> > >
> > > **Seed 1:** 0.815 ± 0.011 (OASIS+LFADS)
> > > **Seed 2:** 0.813 ± 0.011 (OASIS+LFADS)
> > > **Seed 3:** 0.829 ± 0.009 (OASIS+LFADS)
> > > **Seed 4:** 0.784 ± 0.009 (OASIS+LFADS)
> > > **Pooled statistics reported:** 0.810 ± 0.005
> > >
> > > We believe that the boost in our CaLFADS results are likely due to the inferred latent factors based on the original random seed being slight outliers in terms of being directly suited for our downstream classification task. The updated results however do appear robust, and show high consistency across seeds. The original results showed that even “less suited” factors still provide improved performance over OASIS+LFADS.
> > >
> > > Of course, it is indeed important to consider both significance and effect size. We believe that the breakdown of results here supports the conclusion that CaLFADS robustly provides an approximate improvement of about 6% as compared to OASIS+LFADS (or also PCA on the dF/F) on our task.
> > >
> > > **Figures showing the real data:** In the updated Appendix J.2, we’ve added the plots of the inferred latent factors, as well as the reconstructed calcium traces and inferred spikes for both CaLFADS and OASIS+LFADS models. The inferred latent factors were used for the classification purpose as shown in Figure 3. Since there is no one-to-one correspondence between the latent factors of the CaLFADS and the OASIS+LFADS models, direct comparison between them is impossible. Though, the smoothness of the inferred factors is an important feature that can be compared across the results of the two models: the inferred latent factors of CaLFADS are smoother and less noisy compared to those of the OASIS+LFADS. However, we believe that the true usefulness of the inferred latent factors is best reflected in the classification results. In the case of acceptance, we will include an improved version of the real data plots in the revised manuscript.
> > >
> > > **ML and empirical results:** We appreciate the concern about the cherry-picking of empirical results in ML to demonstrate that an algorithm provides an improvement on a task. We can assure you that that has not been the approach here, particularly as the authors all have experience directly both in calcium imaging and electrophysiology data analysis. Instead, the primary goal was to take a tool (LFADS) that has proven a very useful tool for the electrophysiology community, and allow it to be applied to its full potential to the booming field of calcium imaging in neuroscience. Regarding the reproducibility concern, we believe that our updated results on different random seeds demonstrate the robustness of our claim with regard to the superiority of CaLFADS compared to other methods.

---

> > > > ### Comment · AnonReviewer1 · 2020-11-25
> > > > **That makes sense, still one nitpick thoug**
> > > >
> > > > Sorry, perhaps this is overly pedantic, but I still don't actually know when you write plus/minus sem, what they are averages of for each seed.
> > > >
> > > > When you pool across seeds, I hope the sem there is the sem over seeds? My math in my head suggests maybe.
> > > >
> > > > I'm also not sure why you are showing SEM vs SD.  The relevant question is whether alg A is better than alg B, in that sense, the relevant notion of variance is SD, not SEM, I would think.  Specifically, if the mean+/-SD intervals across algorithms overall, there is weak evidence in support of one over the other.
> > > >
> > > > It seems to me that your mean-SD for CaLFADS is bigger than your mean+SD for OASIS+LFADS,
> > > > but when extending to 2SD to correspond to about 95% confidence intervals, they may overlap.
> > > > In other words, I think a t-test p-value would be about 0.05 +/- epsilon.
> > > > Assuming the trials are matched, however, a paired t-test would be significant.
> > > >
> > > > To be clear, I think doing the comparison correctly matters, the actual magnitude of p-value does not matter to me. I *think* what I wrote above is correct, though if you think I'm confused about something, please let me know.

---

> > > > > ### Author Response · Authors · 2020-11-25
> > > > > **Response about standard deviations vs standard error of the mean**
> > > > >
> > > > > Not at all, we are happy to clarify. In drafting a response, it has come to my attention that the term standard deviation suffers from ambiguous usage.
> > > > >
> > > > > Here, we have been using the expression **standard deviation (SD or sigma)** to refer to the **sample standard deviation**, calculated approximately as the **square root of the mean squared error**.
> > > > >
> > > > > We use the expression **standard error of the mean (SEM)** for the **SD/sqrt(n)** where n is the sample size.
> > > > >
> > > > > Unfortunately, it seems that the term standard deviation is used in some cases as a generic term to designate the measure of **the spread of a statistic** of a sampling distribution. Therefore the SEM may be referred to in some places as the **standard deviation of the mean**, instead of reserving the expression standard deviation for the **standard deviation of a distribution**. For example, in several references discussing how one estimates a 95% CI, the term standard deviation is used to designate the SEM as defined above. (This is made explicit right below equation 2.6 on p. 8 of http://www.mit.edu/~6.s085/notes/lecture2.pdf, and near the bottom of the page, you can see that in the confidence interval estimation, the value labelled as "std. dev" is in fact the SEM, as defined above).
> > > > >
> > > > > We apologize for not noticing earlier that this could be a source of confusion. Returning to our submission, using the terminology defined above, in Fig. 3, **we plot the mean +/- SEM across all classifiers pooled across seeds**. For example, since we run 50 classifiers per seed, and 4 seeds were run, the standard error of the mean is computed across 200 classifiers for OASIS+LFADS and across 200 classifiers for CaLFADS.
> > > > >
> > > > > In estimating the results of the independent t-test, using the SEM (SD/sqrt(n)) as in the definition of confidence interval estimations:
> > > > > **Upper** bound of the 95% CI over the **mean OASIS+LFADS recall** ~= 0.810 + 2 * 0.005 = **0.820**
> > > > > **Lower** bound of the 95% CI over the **mean CaLFADS recall** ~= 0.871 - 2 * 0.004 = **0.863**.
> > > > >
> > > > > This estimation supports our more rigorous t-test finding (where we use a Bonferroni corrected p-value threshold, and therefore a wider CI) that the two groups have significantly different means.
> > > > >
> > > > > **An additional note on SDs**
> > > > > We would like to note that we agree that plotting the SDs for each group also provides interesting information, specifically about the spread of the distribution in classifier recall. However, we contend that plotting the SEM (which in our experience is very typical for this type of data), is more helpful in estimating visually whether the means of two groups are significantly different from one another.
> > > > >
> > > > > Lastly, as mentioned, the standard deviations can be retrieved by multiplying the SEMs by sqrt(200) ~= 14.15 for OASIS+LFADS and CaLFADS. Thus, for the pooled results, we get
> > > > > **Mean +/- SD across classifiers:**
> > > > > OASIS+LFADS: 0.810 +/- 0.071
> > > > > CaLFADS: 0.871 +/- 0.057
> > > > >
> > > > > As you can see, both methods in fact yield similar standard deviations.

---

### Official Review · AnonReviewer3 · 2020-10-28
**The authors introduce a novel extension of hierarchical VAEs to sequential data where in this work they focus on data obtained from calcium imaging. While the proposed approach is great and the results are very promising, the structure of the paper needs work as many details are not explained well or missing.**

**Rating:** 7
**Confidence:** 4

**Review:**

##########################################################################

Summary:

Traditionally, the problem of disentangling neural activity from calcium dynamics *and* inferring latent variables from neural activity have been treated separately; the neural activity is first inferred from calcium traces (using a variety of different approaches) and then data analysis is performed on the inferred neural activity. Instead, the authors tackle both of these problems **jointly** by learning a hierarchy of dynamical systems where one is responsible for modeling the calcium dynamics which is driven by another latent system that is responsible for modeling the low-D latent dynamics underlying the population activity; the approach is titled CaLFADS which is a novel extension to both LFADS and variational ladder autoencoder (VLAE) to sequential data.

##########################################################################

Pros:

I thought the proposed approach was super cool! To the best of my knowledge, this is the first time I have seen a hierarchical VAE applied to sequential data which is very exciting and very non-trivial. The empirical results presented on both synthetic and real data are also great as the authors demonstrate the ability to disentangle the neural factor from the calcium dynamics.

##########################################################################

Cons:

My biggest con is that the paper is not clear enough. As I mentioned previously, this work is a non-trivial extension to not only LFADS but **also** to hierarchical VAEs. Sadly, not enough information was given explaining the details. First, the authors should have at least provided a brief introduction to variational ladder autoencoders.

Second, the mathematical form of the generative model does not appear anywhere in the manuscript. Instead, it is explained using figure 1 and the accompanying paragraphs (the first two paragraphs in section 2) but in my opinion, these explanations are opaque without equations accompanying them. For instance, in the original LFADS paper all the variables were a deterministic transformation of the latent variables (the initial condition and the inferred inputs); this allows the system to be completely characterized by these two groups of latent variables. From reading the two paragraphs in section 2, I thought this was also the case for CaLFADS but once I looked at the algorithm table in the appendix I found that $g_{1,t}\sim \mathcal{N}(\mu_{g_{1,t}}, \sigma^2_{g_{1,t}})$! (Note that this stochastic mapping prevents the system to be completely characterized by just the initial conditions and the inferred inputs as now there is uncertainty over $g_{1, t}$ which effectively makes $g_{1, t}$ a latent variable.)

Last, I am still confused about the functional form of the likelihood relating fluorescence to the latent calcium dynamics. In algorithm table 1 line 28 it says  $x_{t} = \textrm{DyeKinetics}(x_{t-1}, z_{1, t})$ but  $\textrm{DyeKinetics}$ is not defined. Towards the end of the second paragraph of section 2, it says “While it is possible to transform calcium dynamics into fluorescence using nonlinear models of fluorescence indicator kinetics (Speiser et al., 2017; Deneux et al., 2016), we found that using the AR(1) process to model the transformation of spike counts to fluorescence was sufficient for accurate reconstruction of the latent space, even in synthetic data with nonlinear calcium transient generation.” Does this mean that $p(x_t \vert z_{1,t}) = \mathcal{N}(Az_{1, t}, \sigma^2)$?

##########################################################################

Questions/Comments:

1) In section 2.1 the authors state that an extra term is added to the loss function that enforces the inferred spike counts, $u_{1, t}$ to follow the dynamics of the higher-order system $z_{2,t}$ through the rate function, $\lambda_t$ i.e. $\mathbb{E}[\log p(u_{1, t} \vert z_{2,t})]$. This seems to be at odds with the VLAE approach but would be enforced if a traditional hierarchical VAE was utilized. Could the authors elucidate why a VLAE with this added regularizer was chosen as opposed to a traditional hierarchical VAE? If a VLAE doesn't work as well without this regularizer it seems like this is a signal that a traditional hierarchical VAE should be used.

2) Just like LFADS, the control inputs are a function of the previous latent factors, thus the variational distribution should depend on the previous control inputs and the initial condition i.e. $Q(u_{2, t} \vert g_{2, 0}, u_{2, < t})$.

3) it would be nice if the authors could show that CaLFADS could do prediction as it would showcase that the model is not only good at inferring the latent factor but is also capable of generating data. One easy way to do this is given N trajectories of length T, 80% of the trajectories are used for training, and then to estimate the other 20% the model generates forward. While I know that the model is evaluated on a held-out test, the initial condition for the system is determined by the full trajectory. Thus, it would be interesting to see how the model works when only a partial length of the trajectory is given.

4) At the bottom of page 16 above equation 10, $gamma$ -> $\gamma$.

5) Many of the figures are not vector graphics.
##########################################################################

To reiterate, I am a big fan of this work and I think it is a very cool contribution. Sadly, more work needs to be done on the structure of the manuscript, specifically on details of the proposed approach. I suggest the authors cut down the intro and use the extra page to really flesh out the details. I will be more than happy to raise my score if the authors do this!

##########################################################################
Post-rebuttal: I have increased my score to a 7. I have laid out my reasoning in response to the author's comments.

---

> ### Author Response · Authors · 2020-11-21
> **Response to reviewer 3**
>
> Thank you for your enthusiasm regarding our manuscript. Unsurprisingly, we also think this is a very cool contribution! We have addressed your core criticisms and we would welcome further feedback to strengthen our paper even more.
>
> **1. The authors should have at least provided a brief introduction to variational ladder autoencoders.** Yes, we agree that a brief intro on VLAEs and hierarchical VAEs would provide useful context and help to highlight our contribution. We have added this in our revision in appendix C.  Please see a draft we have prepared here: https://pdfhost.io/v/gdy6BjPtq_CaLFADS_new_pagespdf.pdf
>
> **2. The mathematical form of the generative model does not appear anywhere in the manuscript.** We apologise for this omission as we felt the pseudocode was sufficient. Clearly it has not been as reviewers 1 and 2 have also remarked on this. We have added the mathematical form of the model in appendix D. Please see a draft we have prepared here:  https://pdfhost.io/v/gdy6BjPtq_CaLFADS_new_pagespdf.pdf
>
> **3. g_1,t is a latent variable and so the system cannot just be described by the initial condition and inferred inputs to the LFADS generator.** g_1,t is indeed a latent variable and relates to the timing of individual spikes observed in that trial. With fluorescence data, it would not be possible to accurately reconstruct the data without a way to infer this timing. As such, the new latent space consists of the initial conditions and inferred input to the generator (as in LFADS), and these variables related to spike-timing. To be able to describe a trial solely in terms of the same latent space as LFADS would require some strategy to marginalize spikes. We are aware of one approach that attempts marginalizing spikes to infer firing rates (Ganmor et al., 2016), but they have no code available to examine their method and it is not obvious how their model could be integrated into a deep neural network architecture. (Please note that there are some notational differences in the revised manuscript. Of particular relevance to this comment is that the old g_1,t is now u_1,t to help clarify that these are unknown inputs to the AR(1) model of calcium dynamics).
>
> **4. Confused about the functional form of the likelihood relating fluorescence to the latent calcium dynamics.** As above, we hope we have clarified this in our revision. To answer your specific concerns, DyeKinetics() is some function relating an AR(1) process to denoised fluorescence transients of which there are many available. As you pointed out, in the linear case P(x_t| z_1,t) = N(Az1,t sigma^2). For simplicity, we just set A=I.
>
> **5. The regularizer added to enforce spike counts follow the dynamics of the higher-order system seems to be at odds with the VLAE approach but would be enforced if a traditional hierarchical VAE was utilized.** We seem to have a different understanding than the reviewer of VLAEs versus VAEs. We understand traditional hierarchical VAEs to have inference networks of the form $z_1 \sim Q_1(z_1|x) = f_1(x)$, $z_2 \sim Q_2(z_2|z_1) = f_2(z_1)$, where $Q_1$, $Q_2$ are the variational posterior distributions, $f_1$, $f_2$ are neural networks. We understand VLAEs to have inference networks of the form $z_1 \sim Q_1(z_1|x) = g(f(x))$, $z_2 \sim Q_2(z_2|x) = h(f(x))$ where $Q_1$, $Q_2$ are variational posteriors, $f$ is a shared neural network for the two posteriors and $h$, $g$ are separate neural networks for the two posteriors. A key part is that expressivity of the network should increases as we go deeper into the hierarchy. If the reviewer disagrees with this characterization we would be interested to hear why.
>
> **6. It would be nice if the authors could show that CaLFADS could do prediction.** As mentioned in an earlier response, the timing of spikes must be inferred to describe a given trial. Since precise spike-timing is not especially predictable yet is the primary source of variation in the signal, it would not make sense to use CaLFADS for prediction, unfortunately.

---

### Official Review · AnonReviewer4 · 2020-10-28
**An extension of a nonlinear method for analyzing the neural dynamics of calcium imaging data**

**Rating:** 5
**Confidence:** 4

**Review:**

This submission extends a previous nonlinear data analysis method ( Latent Factor Analysis of Dynamical Systems, LFADS) to deal with the calcium imaging data (CaLFADS).

Quality: Developing appropriate methods to handle the calcium imaging data is an important question as calcium imaging becomes popular. This submission represents one such attempt. Although some of the results are promising, unfortunately the work is rather incremental. Furthermore, the robustness of the method (as well as the comparisons to alternative methods)  under more realistic noise regime is not carefully tested. Also, the improvement over simple baseline model is small.
Clarity: The writing is overall good, although the paper would benefit by more clearly describing and motivating the assumptions/approximations of the algorithm ( see later for a few more detailed comments).
Originality: The method for handling the calcium traces is novel, although it is a relatively straightforward modification of LFADS.
Significance:  As there is only small improvement over simpler, alternative methods, I feel the significance is relatively low. Also, no new insights were shown by applying these methods compared to simple alternatives.
Pros:
1. CaLFADS is a holistic approach to infer the latent dynamics from fluorescence traces and avoids the two-step procedure which is nice.
2. The method can obtain measures of the uncertainty- an issue that was ignored in most of the previous work on calcium data analysis.
3. The method shows some improvement over a few alternative methods when fitting the data in certain regimes.
Cons:
1. The work is rather incremental. The improvement over the alternative models is small.  This could be seen in Table 1, Fig S2, S3 and Fig. 3 . In Table 1, the margin between the different modes is small. I worry that, in most scientific applications, the small improvement of CaLFADS over Gaussian-LFADS and OASIS-LFADS is unlikely to matter. Take the Lorenz Attractor for example. If the 3-d state-space reconstruction is plotted, the basic structure of the Lorenz Attractor will be recovered for all three models based on other results shown there. It appears that no additional insights are revealed by CaLFADS.
2. The robustness of the method regarding parameter mis-specification and realistic measurement noise is not evaluated. The simulated calcium traces shown in the figures suggest that authors were focusing a high SNR regime- a regime that is very rare practically for large-scale calcium imaging. For example, real fluorescence traces when imaging hundreds of neurons are much noisier than the traces shown in Figure S2. For this to be more convincing for real applications, I would strongly suggest the authors to increase the noise when simulating the fluorescence traces, and test and compare the different models. I am concern that the results may change dramatically in the more realistic noise regime.
3. It is unclear whether OASIS-LFADS is worse than CaLFADS  is simply due that OASIS-LFADS method did not carefully model the noise characteristics of the deconvovled spikes from calcium. For the CaLFADS, the authors mentioned that they used a continuous relaxation to transient amplitudes inspired by a zero-inflated gamma model by Wei et al 2019. While whether this additional ingredient helped with the performance of CaLFADS is unclear, I wonder if the authors had similarly incorporated this zero-inflated gamma model to the OASIS-LFADS method to make it a more fair comparison.
4. Another potential concern is that, practically, it might be difficult to use as the proposed method is complicated and seems to require a fair amount of hyper-parameter tuning. If methods like PCA could already reveal the key insight from a dataset (e.g., Fig. 3), one may question how much value this complicated nonlinear method adds.

More specific comments:
* How well does the Gaussian-LFADS method perform on the real data (i.e., mouse primary visual cortex in Sect. 3.3)? Is it significantly worse than CaLFADS? I am puzzled why this is not reported given Gaussian-LFADS  is one of the main alternative models used in the simulated data. Also, it seems that CaLFADS only slightly outperforms PCA (which is a very basic baseline) for the real data, which again is concerning.
* The parameter of the AR(1) is inferred from a separate model (Friedrich et al 2017). It is a bit disappointing that the model could not infer these parameters directly, as one would think that modeling the underlying dynamics of the computation would further constrained the AR(1) model and help with the deconvolution process (I had considered this as one potential advantage of this holistic approach, but unfortunately it didn’t quite work out). How much do the results depend on the accuracy of these parameters? This is relevant, because practically Friedrich et al 2017 may not perform particularly well in recovering the AR(1) parameters when perform large-scale imaging. Thus it would be useful to check the robustness of the proposed method.
* Related to the last point- when fitting the simulated data, do the authors use the ground truth parameters from the AR(1) model? If that’s the case, it would be useful to test if the model is robust to the model mis-specification, and check if the CaLFADS model still output alternatives in this regime.

Clarification questions:
* “The approximation results from the use of an L1 norm on the spikes,…a range that is poorly handled by the log-gamma function. ” This is a bit confusing. What “data” were referred to here? Inferred “spikes” from calcium traces or the spike train data from electrophysiology?
* The variables in Fig1 Panel B are not explained in the figure caption.
The following paper is relevant as they also proposed a holistic approach to infer the underlying neural structure directly from the raw fluorescence trace.   Aitchison, L., Russell, L., Packer, A. M., Yan, J., Castonguay, P., Hausser, M., & Turaga, S. C. (2017). Model-based Bayesian inference of neural activity and connectivity from all-optical interrogation of a neural circuit. In Advances in Neural Information Processing Systems (pp. 3486-3495).
*The paper by Hernandez et al (2020) is also related to this paper. They proposed methods to model the neural dynamics based on the ephys and calcium data.    Daniel Hernandez et al., Nonlinear Evolution via Spatially-Dependent Linear Dynamics for Electrophysiology and Calcium Data, the NBDT journal, 2020.
* “it would ignore the population dynamics that could be informative for separating out the calcium dynamics from the higher-order computational dynamics. ”  It is unclear what this means. Please clarify the idea here, in particular what the high-order computational dynamics means. There also seems be an issue of timescale here, which the paper didn’t touch on. If the neural dynamics is much slower than the calcium dynamics, it does not seem to matter to much in terms of whether calcium dynamics is carefully modeled.
* The last section before Section 2.1, “The hidden state of this GRU is…to provide an approximation to spike counts”. This needs to better explained.

%%%% after the rebuttal:

I would like to thank for the authors for their effort to address my concerns. The manuscript is now improved, and I am raising my score to 5.  After considering other discussions/comments, overall I still think the manuscript is below the threshold.

---

> ### Author Response · Authors · 2020-11-21
> **Response to reviewer 4 (part 1)**
>
> Thank you for writing such a thorough review. Your review highlighted many ways in which we could improve our paper. Here we address your concerns.
>
> **1. The improvement over the alternative methods is small.** We respectfully disagree with Reviewer 4’s primary justification for our low score, namely that the performance gain offered by CaLFADS is insufficient. We would like to stress the point that the improvement in performance by CaLFADS over other methods is statistically significant. If one’s objective is only to recover the general topology rather than the true attractor manifold, then CaLFADS may offer little over other methods. However, this is also the case for LFADS, as noted in Fig. S2 of the original LFADS paper (Pandarinath et al, 2018) which shows that it is not necessary for recovering the general topology. If this is one’s aim, shallower ML methods are all that is required e.g., see vGLP (Zhao and Park, 2017, https://github.com/catniplab/vlgp/blob/master/notebook/lorenz.ipynb). What we show instead is that when we improve the quality of the latent space, we can significantly improve decoding performance on a downstream task - in this case, the classification of surprise trials from V1 data. Indeed, the latent factors inferred by CaLFADS lead to a statistically significant improvement in surprise trial recall over latent factors from alternative methods. Furthermore, to reinforce our claim that CaLFADS learns more meaningful latent factors, we retrained all models with multiple seeds. To our surprise and fortune, the performance gain only increased (CaLFADS: 0.871, OASIS+LFADS: .810, Gaussian-LFADS: 0.500, PCA: .815). We believe this is a valuable improvement over previous methods, and would like further details on what level of improvement Reviewer 4 would consider sufficient for acceptance.
>
> **2. The robustness of the methods regarding parameter mis-specification is not evaluated.** We do in fact include a scenario where the parameters are mis-specified. Indeed, the results that were reported in the manuscript are for the case where we infer the AR(1) parameters (decay \tau, spike amplitude A and emissions noise \sigma)  using OASIS, as one would with real data. As such, this is a realistic scenario in which parameters are mis-specified. We have also compared the performance of OASIS+LFADS and CaLFADS in the linear calcium model case where the ground-truth AR(1) parameters (\tau=0.3, A=1.0, \sigma=0.25) are given to OASIS to extract spikes, and given to CaLFADS for fluorescence reconstruction. In this scenario where parameters are perfectly specified, OASIS+LFADS is better at reconstructing the Lorenz attractor (R-sq = .977 vs.956, statistically significant). Since CaLFADS is better than OASIS+LFADS at reconstructing the Lorenz attractor in the mis-specified case, this indicates that CaLFADS is more robust to parameter mis-specification. Furthermore, with synthetic data generated with a nonlinear calcium model, the AR(1) process is still used to reconstruct fluorescence traces. In this case, both the model and parameters are mis-specified. As reported in the paper, the performance of CaLFADS over OASIS+LFADS is even more pronounced, demonstrating that CaLFADS is more robust to mis-specification than OASIS+LFADS.
>
> **3. The robustness of the methods regarding realistic measurement noise is not evaluated.** Would it be possible to be more specific about what is meant by a realistic measurement noise? We set the change in fluorescence from a single spike to be 4x the standard deviation of the emissions noise, which is comparable to other calcium fluorescence simulations.
>
> **4. If the authors had incorporated this zero-inflated gamma model to the OASIS-LFADS method to make it a more fair comparison.** The zero-inflated gamma model is based on the static distribution of fluorescence amplitudes from OASIS. As such, it is already incorporated into the OASIS+LFADS method. We will make this clearer in our revision.
>
> **5. It might be difficult to use as the proposed method is complicated and seems to require a fair amount of hyper-parameter tuning.**  It’s true that it takes longer to train CaLFADS than OASIS+LFADS (about 3x longer). However, it did not require extensive hyperparameter tuning. We tried a couple of sets of different layer widths and KL annealing factors, but these did not impact the results. We were in fact able to run all experiments reported in the paper on a 3-year old GPU-enabled Thinkpad, and therefore we believe other researchers should find the method entirely feasible.

---

> > ### Author Response · Authors · 2020-11-21
> > **Response to reviewer 4 (part 2)**
> >
> > **6. It is disappointing that the model could not infer AR(1) parameters directly.** It is true that CaLFADS does not infer AR(1) parameters directly. We tried using CaLFADS to fine-tune AR(1) parameters inferred by OASIS, but this made Lorenz attractor reconstruction worse. Intuitively, we think this is because the reconstruction loss is extremely sensitive to AR(1) parameter changes during training and so gradients are dominated by changes to the loss wrt these parameters. Since we could obtain very good results by keeping AR(1) parameters fixed, and we thought these results were interesting in their own right we decided to leave this problem to future research.
> >
> > **7. Related Work**: We appreciate Reviewer 4 pointing out related work. We note the similarities of our generative model to Aitchison et al (2017). A key difference here is that we try to find nonlinear latent dynamics, whereas their model comprises a linear dynamical system in the latent space. Furthermore, theirs is a fully Bayesian approach, whereas CaLFADS has many more deterministic components. We also note the capabilities of VIND (Hernandez et al., 2020) at inferring non-linear latent dynamics in wide-field calcium imaging data. Although we’re very enthusiastic about this approach and its ability to be applied across different scales of neural data (single cell electrophysiology and wide-field calcium imaging) we note that it is not attempting to separate sets of latent dynamics. Indeed, a notable omission from the paper is the application to two-photon calcium imaging where it would be necessary to disentangle calcium dynamics from population dynamics.
> >
> > **8. “it would ignore the population dynamics that could be informative for separating out the calcium dynamics from the higher-order computational dynamics. ” It is unclear what this means.** We apologise for the lack of clarity here. We note that other reviewers also found the ‘higher-order’ and ‘lower-order’ dynamics terminology confusing. We propose to change this terminology to ‘deeper-level’ and ‘shallower-level’ dynamics respectively. The background to this is the idea that neural dynamics possess some low-dimensional structure that enables a population to perform some computation. As such, what we mean by this statement is that ignoring correlations in population activity can harm the ability to separate calcium dynamics (which is independent of population activity) and computational dynamics (which is not independent of population activity).
> >
> > **9. If the neural dynamics is much slower than the calcium dynamics, it does not seem to matter too much in terms of whether calcium dynamics is carefully modeled.** It is for precisely the reason that calcium dynamics and underlying computational dynamics operate on similar timescales that we designed CaLFADs to be able to separate them. There may of course be cases where experimentalists are studying neural activity associated with cognitive or behavioural tasks that are much slower than calcium dynamics (time-scales of seconds, rather than 100s of ms), e.g., long term memory tasks. CaLFADS would not be well suited for studying such data for two reasons. The first reason, as the reviewer pointed out, is that there would be no need to separate time scales of dynamics. The second reason is that the time scale of computational dynamics would be too long to collect sufficient trials required to train a model like LFADS or CaLFADS. For this reason, like LFADS, CaLFADS is best suited to studying computational dynamics in sensorimotor and perceptual decision making tasks where it is possible to collect vast numbers of trials. Given that the majority of 2-photon calcium imaging data is collected with these kinds of tasks, we feel that CaLFADS helps to solve a very pressing problem.
> >
> > **10. No comparison with Gaussian-LFADS on the mouse V1 data.** We have now run this experiment and found that Gaussian-LFADS performs no better than chance (average recall = 0.5), indicating that this model can not separate the calcium and computational dynamics sufficiently to encode surprise in the latent factors.

---

> > ### Comment · AnonReviewer4 · 2020-11-25
> > **more clarification questions**
> >
> > I appreciate the authors spending time addressing my questions. A few more clarification questions:
> > 1. In the response, the authors stated " The zero-inflated gamma model is based on the static distribution of fluorescence amplitudes from OASIS. As such, it is already incorporated into the OASIS+LFADS method. We will make this clearer in our revision." This response is puzzling to me. Did the authors mean that the OASIS+LFADS method in fact  already explicitly incorporated a zero-inflated gamma model for the LFADS step? I thought the authors did OASIS first, then perform LFADS without explicitly modeling the detailed noise characteristics of the "deconvolved spikes". What noise model is used there?
> > 2. I still think it would be informative to show and compare the 3-d state-space reconstructions for the Lorenz Attractor. I'd be more convinced if there is anything new revealed by CaLFADS methods compared to the alternatives there by examining the structure of the latent.
> > 3. It's shocking to see the authors' statement that Gaussian-LFADS is at exactly at chance level for the V1 data (0.5), while simple method such as PCA already achieves decent performance there. That seems to be an extraordinary statement to me. Maybe it is due to some profound mis-specification of the hyper-parameters? Are the authors confident about that result?

---

### Author Response · Authors · 2020-11-21
**Thank you to all reviewers**

We thank the reviewers for their assessment of our paper. While we feel that our scores were harsh, we have noted several areas for improvement. We are working on completing the revised manuscript for submission before the rebuttal deadline. For now, please see additional material we are proposing to include in the appendix.

https://pdfhost.io/v/gdy6BjPtq_CaLFADS_new_pagespdf.pdf

**1. Related Work and contribution.** Here we discuss LFADS, and an assortment of variational autoencoders that have been developed to solve specific tasks when analysing calcium imaging data including DeepSpike, VIND, LeMoNADe. We illustrate here the methodological novelty of our approach and show how CaLFADS is able to solve a problem that no other method is capable of.

**2. Introduction to Variational Ladder Autoencoders.**  Here we give a brief outline of the theory of variational ladder autoencoders and the problem they are intended to solve, namely the ability to use expressiveness at deepening levels of the network hierarchy to control the abstractness of features encoded in the latent space. The VLAE architecture was well suited to our problem, where we wanted to separate the latent representation of computational dynamics (more abstract) from calcium dynamics (less abstract). We recommend that our reviewers read the original paper for a more thorough explanation (https://arxiv.org/abs/1702.08396).

**3. Equations for the model.** 3 reviewers would have liked to have seen equations for the full model of CaLFADS. We have now written these up. In doing so, we have adjusted the notation slightly and we will be checking thoroughly for consistency throughout the rest of the manuscript before the final rebuttal deadline.

Additionally, to reinforce our claims about the performance improvements of CaLFADS we retrained all models using multiple seeds on the mouse V1 data. We found that the performance gain _widened_ (**CaLFADS: 0.871**, OASIS+LFADS: .810, Gaussian-LFADS: 0.500, PCA: .815). The only changes that have been made are newer hardware and more recent versions of Pytorch.

---

### Author Response · Authors · 2020-11-23
**Revision uploaded**

Thank you again for taking the time to review our paper in detail. Please find our revision which includes all of the amendments listed below.

---

### Author Response · Authors · 2020-11-25
**Second revision uploaded**

Please find a second revision uploaded with the following amendments:

**Figures showing fluorescence traces, spikes and latent factors** were added for both CaLFADS and OASIS+LFADS applied to the mouse V1 data **(Appendix J2)**

We clarified what we had in mind, at the **very end of the introduction**, when we said that **CaLFADS has the potential to be applied** to different domains in the life sciences. Specifically, we describe its potential application to fMRI data and even more broadly to the modeling of data types like insurance claims, given the shared hierarchical dynamical structures of these datatypes.

Thank you again for taking the time to review our submission and to provide feedback that has greatly helped us improve its quality and clarity.

---

### Decision · Program_Chairs · 2021-01-07
**Final Decision**

**Decision:**

Reject

**Comment:**

Inferring latent trajectory from noisy Ca time series is an important and timely problem and the current study shows some progress in the inference problem. Although the proposed model has some originality, there are remaining issues rendering the manuscript not ready for publication yet. Reviewers raised issues on readability, lack of details, statistics of experiments, Ca time constant estimation, effect size, and lack of comparison. Through an extensive discussion and revisions, I'm happy to see the manuscript was greatly improved in readability and additional statistics were provided. However, the Gaussian-LFADS' performance at exactly chance raises red flags, effect size is small, and the significance of the scientific findings remain weak. The model is presented as a variational ladder autoencoder system with 2 layers, but the shallow latent representation is tied to the continuous approximation of the point process likelihood. Hence, I view the model as an extension of LFADS rather than a flat-hierarchical VAE.

Overall, this paper has a potential of becoming a solid contribution for statistical neuroscience, once the above shortcomings are addressed.